# The ALPK1/TIFA/NF-κB axis links a bacterial carcinogen to R-loop-induced replication stress

Michael Bauer[1,11], Zuzana Nascakova[2,3,11], Anca-Irina Mihai [1,11], Phil F. Cheng [4], Mitchell P. Levesque[4], Simon Lampart[1], Robert Hurwitz[5], Lennart Pfannkuch[5], Jana Dobrovolna[2], Melanie Jacobs[6], Sina Bartfeld[6], Anders Dohlman[7], Xiling Shen [7], Alevtina A. Gall[8], Nina R. Salama[8], Antonia Töpfer[9], Achim Weber [1,9,10], Thomas F. Meyer[5], Pavel Janscak [1,2,10,12✉] & Anne Müller [1,10,12✉]

Exposure of gastric epithelial cells to the bacterial carcinogen *Helicobacter pylori* causes DNA double strand breaks. Here, we show that *H. pylori*-induced DNA damage occurs co-transcriptionally in S-phase cells that activate NF-κB signaling upon innate immune recognition of the lipopolysaccharide biosynthetic intermediate β-ADP-heptose by the ALPK1/TIFA signaling pathway. DNA damage depends on the bi-functional RfaE enzyme and the Cag pathogenicity island of *H. pylori*, is accompanied by replication fork stalling and can be observed also in primary cells derived from gastric organoids. Importantly, *H. pylori*-induced replication stress and DNA damage depend on the presence of co-transcriptional RNA/DNA hybrids (R-loops) that form in infected cells during S-phase as a consequence of β-ADP-heptose/ ALPK1/TIFA/NF-κB signaling. *H. pylori* resides in close proximity to S-phase cells in the gastric mucosa of gastritis patients. Taken together, our results link bacterial infection and NF-κB-driven innate immune responses to R-loop-dependent replication stress and DNA damage.

[1] Institute of Molecular Cancer Research, University of Zurich, 8057 Zurich, Switzerland. [2] Institute of Molecular Genetics, Academy of Sciences of the Czech Republic, 142 20 Prague, Czech Republic. [3] Faculty of Science, Charles University in Prague, 128 00 Prague, Czech Republic. [4] Department of Dermatology, University Hospital Zurich, Zurich, Switzerland. [5] Max Planck Institute for Infection Biology, Department of Molecular Biology, 10117 Berlin, Germany. [6] Research Center for Infectious Diseases, Institute for Molecular Infection Biology, University of Würzburg, 97080 Würzburg, Germany. [7] Biomedical Engineering, Duke University, Durham, NC, USA. [8] Division of Human Biology, Fred Hutchinson Cancer Research Center, Seattle, WA, USA. [9] Department of Pathology and Molecular Pathology, University Hospital Zurich and University of Zurich, Zurich, Switzerland. [10] Comprehensive Cancer Center Zurich, Zurich, Switzerland. [11]These authors contributed equally: Michael Bauer, Zuzana Nascakova, Anca-Irina Mihai. [12]These authors jointly supervised this work: Pavel Janscak, Anne Müller. ✉email: pjanscak@imcr.uzh.ch; mueller@imcr.uzh.ch

Observational studies in humans first described a link between infection with the human gastric pathogen *Helicobacter pylori* and gastric cancer in the early 1990's[1,2]. This discovery led to the categorization of *H. pylori* as a class I (highest class) carcinogen by the World Health Organization in 1994. A causal link between *H. pylori* infection and gastric adenocarcinoma was later confirmed in experimental models using Mongolian gerbils[3] and inbred mouse strains[4,5], especially in settings of hypergastrinemia[6] or a high salt diet[7]. Gastric *H. pylori*-induced carcinogenesis is preceded by a series of precursor lesions that manifest as chronic inflammation, atrophy, intestinal metaplasia, and dysplasia[8]. The *H. pylori* type IV secretion system (T4SS), which is encoded by the Cag pathogenicity island (Cag-PAI), and its T4SS-translocated effector CagA have been described as major risk factors of gastric cancer and its precursor lesions in observational studies in humans[9] and in experimental models[10,11]. As a consequence of a large body of evidence implicating *H. pylori* in gastric carcinogenesis, screening programs using upper gastrointestinal tract endoscopy are now in place in countries with a particularly high gastric cancer burden, such as South Korea or Japan. These programs have allowed for early detection of gastric cancer and have reduced mortality from this disease[12]. Eradication therapy targeting *H. pylori* is efficacious at reducing gastric cancer risk, especially if it is applied in patients with non-atrophic or atrophic gastritis but without evidence of metaplasia[13]. A recent study has shown that even patients with early gastric cancers that are limited to the gastric mucosa or submucosa can benefit from eradication therapy, as they show a lower risk of progressing to metachronous gastric cancer than non-eradicated controls[14].

We and others have demonstrated in several independent studies that *H. pylori* induces DNA double-strand breaks (DNA DSBs) in gastric epithelial cells in vitro and in vivo[15–17]. DNA DSB induction in *H. pylori*-exposed cells depends on a functional T4SS[16–18] and preferentially occurs in transcribed regions of the genome[17]. Whereas translocation of the only described protein substrate of the T4SS, CagA, does not contribute to DNA DSB induction, the depletion of NF-κB subunits strongly reduces DNA DSB formation, suggesting that *H. pylori*-induced DNA damage is driven by active transcription of NF-κB target genes, which in turn is Cag-PAI-dependent[18]. Interestingly, a reduction in the level of *H. pylori*-induced DSBs is also observed upon depletion of the nucleotide excision repair endonucleases XPG and XPF[18] that have been shown to cleave co-transcriptional RNA/DNA hybrids, termed R-loops[19,20].

The earliest immune response to *H. pylori* is initiated by gastric epithelial cells, to which *H. pylori* adheres tightly in vitro and in vivo. Gastric epithelial cells sense *H. pylori* through a recently described innate immune defense pathway that is comprised of a sensor, the alpha-kinase 1 (ALPK1), and a signaling adaptor, the tumor necrosis factor receptor-associated factor (TRAF)-interacting protein with forkhead-associated domain (TIFA), which together initiate T4SS-dependent NF-κB signaling[21–24]. The pathway is activated upon T4SS-dependent delivery of either D-glycero-beta-D-manno-heptose 1,7-bisphosphate (HBP) or ADP-beta-D-manno-heptose (β-ADP-heptose) into the host cell[21–24]; both molecules are metabolic precursors of lipopolysaccharide (LPS) biosynthesis. The binding of β-ADP-heptose, and possibly of β-ADP-heptose 7-P (generated in the host cell from HBP), to ALPK1 stimulates its kinase domain to phosphorylate and activate TIFA[24,25], which forms large complexes (TIFAsomes) that also include interactors such as TRAF2[22]. *H. pylori* mutants that lack the ability to produce HBP or β-ADP-heptose are incapable of activating the ALPK1/TIFA pathway[22,23]; conversely, the extracellular addition of β-ADP-heptose alone is sufficient to activate NF-κB signaling in

an ALPK1/TIFA-dependent manner[24]. Other Gram-negative pathogens are sensed via the same pathway; these include *Salmonella typhimurium*, *Shigella flexneri*[26], *Yersinia pseudotuberculosis*[25] as well as *Neisseria meningitidis* and *Neisseria gonorrhoeae*, for which the pathway was first described[27]. In the case of all these pathogens, the activation of the ALPK1/TIFA signaling axis lead to NF-κB activation and the subsequent production of pro-inflammatory cytokines and chemokines, of which the most often investigated example was IL-8.

As DNA damage induced by *H. pylori* is dependent on NF-κB activation, and NF-κB activation is now known to be triggered by the ADP-heptose/ALPK1/TIFA axis, we set out here to investigate a possible link between the two processes and to elucidate in detail the molecular mechanism of *H. pylori*-induced DNA damage.

## Results

**H. pylori induces ALPK1/TIFA/NF-κB-dependent DNA damage.** To investigate whether the β-ADP-heptose/ALPK1/TIFA signaling axis is involved in *H. pylori*-induced DNA damage, we took advantage of AGS gastric epithelial cells that were either proficient or deficient for ALPK1 and TIFA expression due to genetic ablation of the respective loci by CRISPR/Cas9 technology[21]. Cells were infected with wild-type *H. pylori* and were subjected to immunofluorescence microscopy to quantify 53BP1 and γH2AX foci, which identify sites of DNA DSBs. In wild-type AGS cells, *H. pylori* exposure induced multiplicity-of-infection (MOI)-dependent DNA damage, which could be observed as early as 6 h post infection (Fig. 1a, b, Supplementary Data 1 and 2). The genetic ablation of *ALPK1* or *TIFA* in AGS cells strongly reduced DNA DSBs as judged by quantification of 53BP1/γH2AX foci (Fig. 1a, b). DNA damage was limited to cells in S-phase, which were identified by PCNA staining (Fig. 1b, Supplementary Fig. 1a, Supplementary Data 2 and 3) and was also observed with a second strain of *H. pylori* (Supplementary Fig. 1b). Plotting the signal intensities of PCNA over DAPI, which readily separates cells in G1, S, and G2/M phases of the cell cycle, confirmed that cells with five and more 53BP1 foci are typically in S-phase (Fig. 1c). To rule out that off-target effects of genome-editing by CRISPR led to the reduction in 53BP1 foci, we took advantage of a second, independently generated ALPK1-deficient AGS line published previously[22] and of *TIFA*-deficient AGS cells that had been complemented by transduction with a lentivirus containing the complete TIFA coding sequence[21]. In these complemented cells, TIFA expression is driven by the lentiviral MND promoter and completely restores IL-8 production upon co-culture with *H. pylori*[21]. When subjected to *H. pylori* exposure followed by 53BP1 staining and the quantification of 53BP1 foci, the second *ALPK1*-deficient cell line phenocopied the effects of the first one (Supplementary Fig. 1c); TIFA-complemented cells exhibited an almost complete restoration of 53BP1 foci formation, whereas the baseline levels (uninfected condition) were unchanged (Supplementary Fig. 1c). These results indicate that the resistance of *ALPK1*- or *TIFA*-deficient AGS cells to *H. pylori*-induced DNA damage is indeed due to *TIFA* ablation and not off-target effects of CRISPR.

A second readout of DNA damage, for which extracted nuclear DNA was separated by pulsed-field gel electrophoresis, confirmed that wild-type AGS cells sustain more DNA DSBs upon *H. pylori* exposure than their ALPK1- or TIFA-deficient counterparts (Fig. 1d, e, Supplementary Data 4). The differential induction of DNA damage as observed in both readouts was mirrored by differential IL-8 production, which, as published previously[21–23], was strongly dependent on ALPK1 and TIFA also in our experiments (Fig. 1f). As the activation of ALPK1/TIFA results in

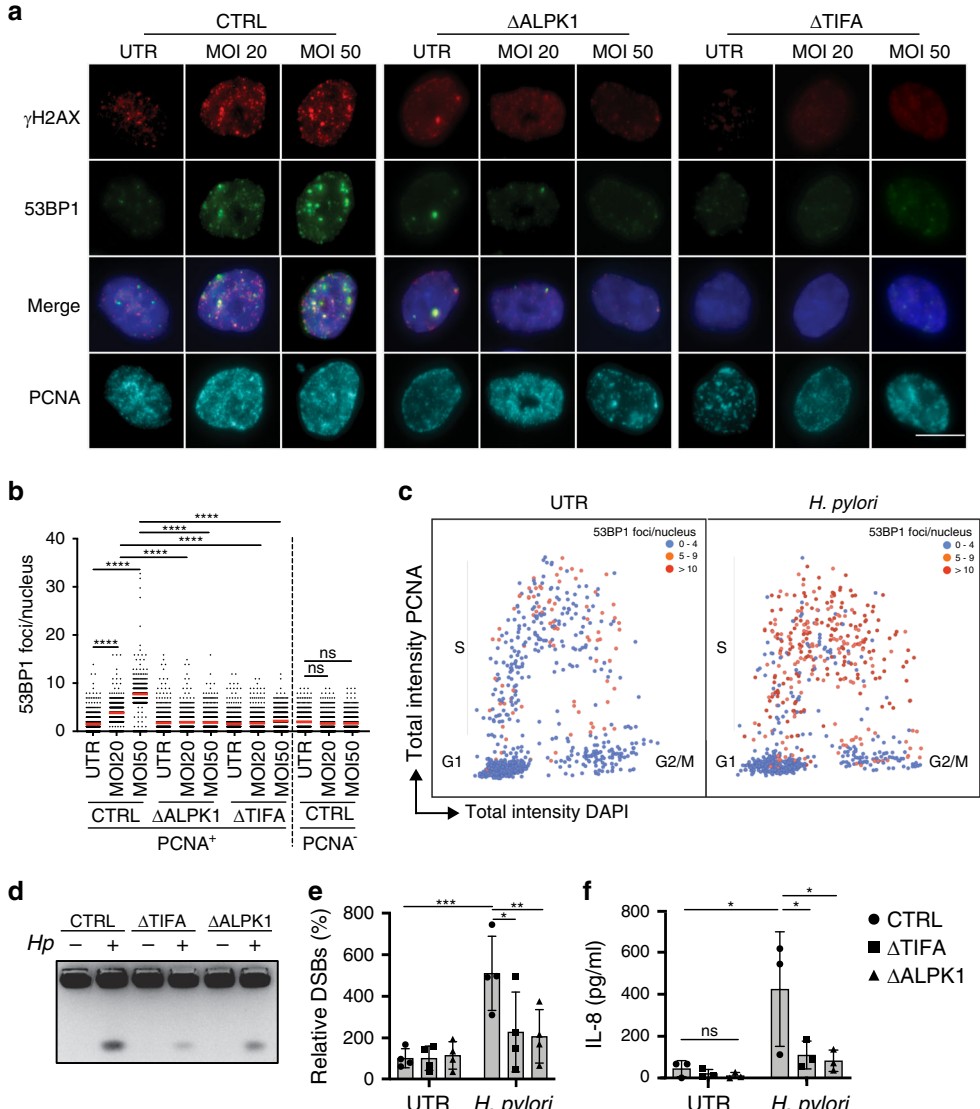

**Fig. 1 *H. pylori* induces DNA DSBs in gastric epithelial cells that depend on ALPK1/TIFA signaling and occur in S-phase. a–c** Wild-type (CTRL), ALPK1-deficient (ΔALPK1), and TIFA-deficient (ΔTIFA) AGS cells were infected for 6 h with *H. pylori* strain P12 at multiplicities of infection (MOIs) of 20 and 50 (or left untreated, UTR) and subjected to immunofluorescence staining for phosphorylated H2AX (γH2AX), 53BP1 and PCNA. DAPI was used to stain nuclei. Representative images are shown in **a** for one PCNA-positive cell per condition, alongside scatter dot plots of the number of 53BP1 foci per nucleus of >553 and up to 650 cells per condition for PCNA⁺ and PCNA⁻ cells of the indicated genotypes in **b**. Data in **b** are pooled from three independent experiments. Red lines indicate medians. In **c**, the PCNA and DAPI signal intensities of ~600 uninfected and as many *H. pylori*-infected (MOI 50) wild-type cells were plotted to visualize cell cycle phase (G1, S, G2/M). The color code indicates the number of 53BP1 foci/nucleus. Scale bar in **a**, 10 μm. **d–f** Wild-type (CTRL), ALPK1-deficient (ΔALPK1), and TIFA-deficient (ΔTIFA) AGS cells were infected for 6 h as described above and subjected to PFGE. A representative gel is shown in **d** alongside the quantification of three gels in **e**. Fragmented DNA was normalized to the intact DNA retained in the slot, and to the uninfected control condition, which was set as 100%. IL-8 secretion as assessed by ELISA is shown in **f**. In **e** and **f**, data are represented as mean +/− SEM of four independent experiments. In **f**, data are represented as mean +/− SEM of three independent experiments. *P*-values were calculated by one-way ANOVA with Dunn's multiple comparisons correction. ns, not significant; *$p < 0.05$, **$p < 0.01$, ***$p < 0.005$, ****$p < 0.0001$.

NF-κB activation, we asked whether the inhibition of NF-κB activation, or generally the inhibition of transcription, would prevent *H. pylori*-induced DNA damage. Indeed, the exposure of cells to an irreversible inhibitor of IKK-α, BAY 11-7082, which prevents phosphorylation of IκBα and the translocation of NF-κB to the nucleus, was as efficient as *ALPK1* or *TIFA* deletion in abrogating the formation of 53BP1 foci in S-phase cells upon *H. pylori* infection (Fig. 2a, b, Supplementary Fig. 2a). Application of the transcription inhibitor triptolide, a natural product extracted from the Chinese plant *Tripterygium wilfordii* that is known to block RNA polymerase II activity[28], had similar effects on DNA damage induction by *H. pylori* as NF-κB inhibition (Fig. 2a, b).

Interestingly, application of the canonical NF-κB activator TNF-α did not result in DNA damage (Fig. 2a, b), even at concentrations that trigger IL-8 secretion at levels comparable to those induced by *H. pylori* infection (Fig. 2c). The consequences of *H. pylori* infection on genomic integrity were comparable in magnitude to DNA damage induced by the topoisomerase inhibitor camptothecin (CPT), the effects of which were however not limited to S-phase cells (Fig. 2a, b, Supplementary Fig. 2a). All results combined indicate that activation of the ALPK1/TIFA/NF-κB signaling axis by *H. pylori* triggers DNA damage in target cells that appears to be specific to this pathway of NF-κB activation, and that requires active transcription and occurs predominantly

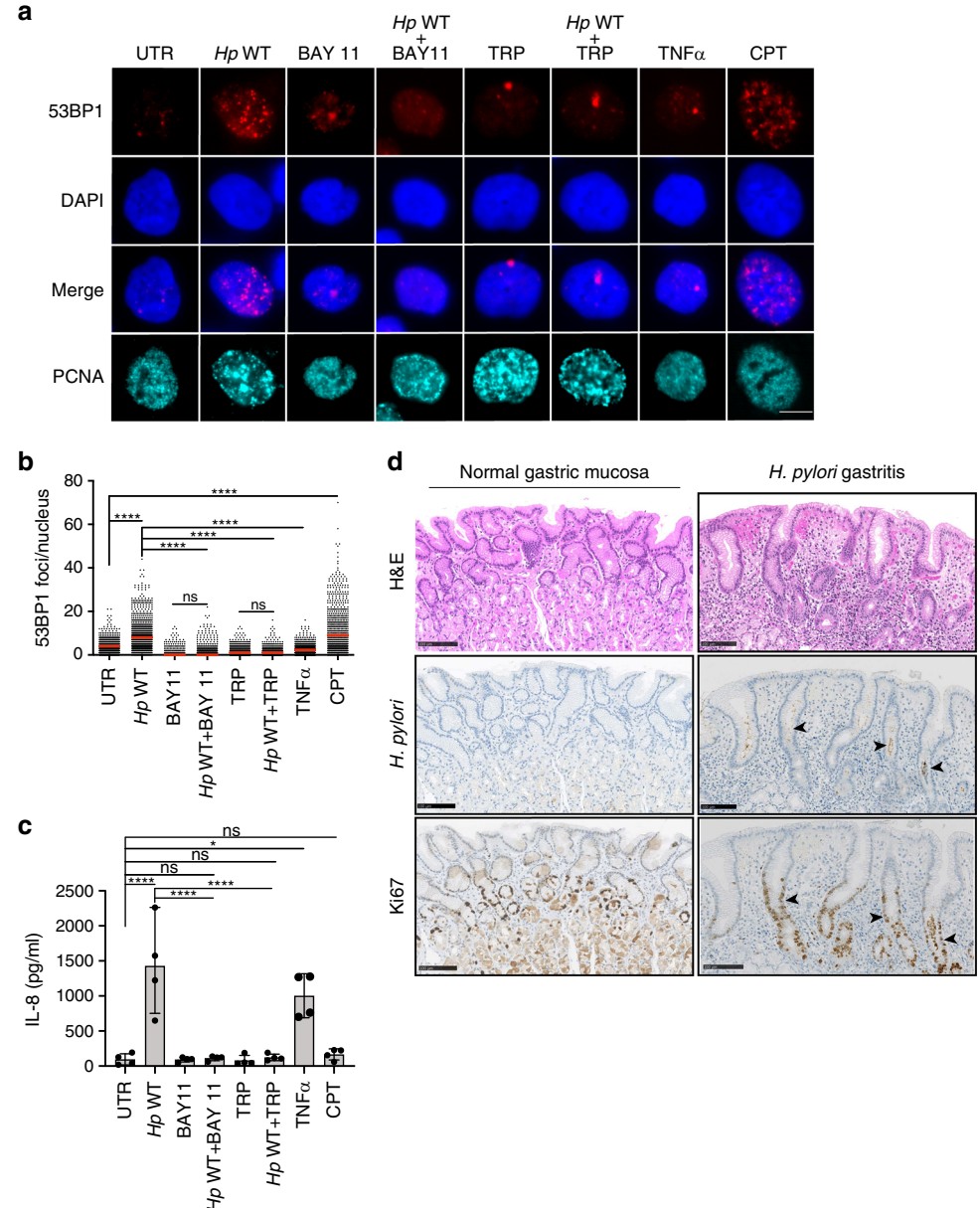

**Fig. 2 H. pylori-induced DNA DSBs require active transcription and NF-κB activation. a–c** Wild-type AGS cells were infected for 6 h with *H. pylori* P12 at an MOI of 50 in the presence or absence of the NF-κB inhibitor BAY 11-7082 and the transcription inhibitor triptolide (1 μM and 100 nM final concentration, respectively), or treated with 10 nM TNF-α or 100 nM camptothecin (CPT) for 6 h, and subjected to immunofluorescence staining for 53BP1 and PCNA. Representative images are shown in **a** (scale bar, 10 μm) alongside scatter dot plots of >430 and up to 954 cells per condition in **b**. IL-8 secretion as assessed by ELISA is shown in **c**. Data in **b** and **c** are pooled from four independent experiments. Red lines indicate medians. Data in **c** are represented as mean + SEM of three independent experiments. *P*-values were calculated by one-way ANOVA with Dunn's multiple comparisons correction in **b** and by two-way ANOVA with Tukey's multiple comparisons correction in **c**. ns, not significant; *p < 0.05, ****p < 0.0001. **d** Immunohistochemical analysis of *H. pylori* staining relative to Ki67+ cells in the gastric mucosa of patients presenting with (right panels) or without gastritis (left panels); six patients with gastritis and three control patients were stained with very similar results. Hematoxylin and eosin (H&E)-stained sections are shown as well. Scale bars, 100 μm. Arrows point to areas where *H. pylori* resides in close proximity to proliferating cells.

in actively replicating (S-phase) cells. As NF-κB activation and the resulting production of reactive oxygen species (ROS), through mechanisms involving inducible nitric oxide synthase (iNOS) and other inflammatory enzymes, have been implicated in *H. pylori*-induced DNA damage[29], we addressed this possibility experimentally using the antioxidant *N*-acetyl-cysteine. We found that *H. pylori* induces ROS, as judged by their flow cytometric detection and quantification (Supplementary Fig. 2b, Supplementary Data 5). However, co-culturing AGS cells with *H. pylori* in the presence of *N*-acetyl-cysteine—at concentrations that

completely abrogated ROS production—failed to reduce the DNA damage as judged by 53BP1 foci formation (Supplementary Fig. 2b–d). This result argues against a major contribution of ROS to *H. pylori*-induced DNA damage in this setting.

To address whether *H. pylori* indeed resides in close proximity to actively replicating cells in the human gastric mucosa, we selected biopsies from six cases of *H. pylori*-associated gastritis and three cases of normal (uninfected) gastric mucosa for immunohistochemical analysis of *H. pylori* and of the proliferation marker Ki67. We indeed found numerous examples of

regions where *H. pylori* could be detected in close proximity to Ki67+ cells in gastric glands; gastric glands typically exhibited more Ki67+ cells than glands of uninfected controls (Fig. 2d). These data indicate that areas of hyperproliferation are readily detectable in the *H. pylori*-infected gastric mucosa, and *H. pylori* resides in close proximity to proliferating cells.

**H. pylori LPS biosynthetic intermediates induce DNA damage.** We next set out to assess which bacterial determinants are required for DNA damage induction in addition to the above-mentioned Cag-PAI/T4SS identified earlier[16–18]. The gene *rfaE* (*hldE*; HP0858) of *H. pylori* encodes a bifunctional enzyme with kinase as well as ADP transferase activity, forms a contiguous operon with *gmhA*, *gmhB*, and *rfaD*, and is involved in the synthesis of both HBP and ADP-heptose as intermediate metabolites of LPS biosynthesis[23]. To examine whether *rfaE* is required for DNA damage induction by *H. pylori*, we infected AGS cells with the wild-type *H. pylori* strain P12 or its *rfaE* and *Cag-PAI* mutants. The *rfaE* mutant induced less DNA damage as determined by 53BP1 immunofluorescence microscopy than the wild-type parental strain, and its phenotype resembled that of the *Cag-PAI* mutant (Fig. 3a, b). Both mutants failed to induce IL-8 production by AGS cells (Fig. 3c); in contrast, the ability of the *rfaE* mutant to deliver CagA and to thereby induce cell elongation and scattering was not compromised (Supplementary Fig. 3a). The reduced DNA damage associated with *rfaE* mutant infection could be confirmed using pulsed-field gel electrophoresis (Fig. 3d, e, Supplementary Data 6) and phenocopied what was previously published for the *Cag-PAI* mutant[18]. As the product of RfaE activity, β-ADP-heptose, has recently been shown to enter host cells and activate ALPK1[24,25], we synthesized the α- and β-anomers of ADP-heptose and determined the consequences of their addition to AGS cells on genomic integrity. The β-anomer, but not the α-anomer, induced DNA damage—as judged by 53BP1 formation—as well as IL-8 secretion in a ALPK1/TIFA-dependent manner (Fig. 3f, g, Supplementary Fig. 3b); which was also confirmed by PFGE of fragmented DNA (Supplementary Fig. 3c, d, Supplementary Data 7). As in the context of live *H. pylori* infection, DNA damage upon β-ADP-heptose exposure was limited to PCNA+ cells (Fig. 3f, Supplementary Fig. 3b, e). The defect of the *rfaE* mutant could not be rescued by addition of TNF-α (Supplementary Fig. 3f). The combined results indicate that *rfaE* activity is required, and its product β-ADP-heptose is sufficient, to induce the ALPK1/TIFA-dependent DNA damage observed upon live *H. pylori* infection of cultured gastric epithelial cells.

We next asked whether the findings could be recapitulated in a gastric organoid model. Gastric organoids were grown from two donors undergoing bariatric surgery[30] and were subsequently transformed into 2D cultures as previously described[31]. Infection of primary cells from gastric organoids with wild-type bacteria, but not the *rfaE* or *Cag-PAI* mutants, resulted in the production and secretion of IL-8 into the culture supernatant (Supplementary Fig. 3g). Exposure of the cells to the β- but not the α-anomer of ADP-heptose elicited a similar IL-8 response as the wild-type *H. pylori* infection (Supplementary Fig. 3g). Importantly, wild-type, but not mutant bacteria-induced 53BP1 foci as judged by immunofluorescence microscopy; as in AGS cells, this increase in 53BP1 focus formation was restricted to S-phase cells identified by PCNA foci (Fig. 3h, i, Supplementary Fig. 3h). The addition of β- but not α-ADP-heptose phenocopied the effect of *H. pylori* infection with respect to 53BP1 focus formation (Fig. 3h, i, Supplementary Fig. 3h). The results from gastric organoid-derived primary cells confirm that DNA damage occurs upon *H. pylori* exposure also in this more physiological setting, is specific

to S-phase, depends on the *H. pylori* enzyme *rfaE* and the Cag-PAI, and can be elicited by the addition of the ALPK1 ligand β-ADP-heptose.

**β-ADP-heptose/ALPK1/TIFA signaling induces replication stress.** Physical obstacles such as active transcription complexes or DNA secondary structures are known to cause the slowing or arrest of replication fork progression, a condition commonly referred to as DNA replication stress[32]. To investigate whether the DNA damage induced by *H. pylori* infection is linked to replication stress, we employed a DNA fiber assay that exploits the ability of eukaryotic cells to incorporate halogenated pyrimidine nucleoside analogs into replicating DNA and provides a powerful tool to monitor genome-wide replication perturbations at single-molecule resolution[33]. Ongoing replication events were sequentially labeled with two thymidine analogs—chlorodeoxyuridine (CldU) and iododeoxyuridine (IdU)—and individual two-color labeled DNA tracts were visualized on stretched DNA fibers by immunofluorescence microscopy (Fig. 4a, b). We found that *H. pylori* infection led to slowing of replication fork progression in wild-type AGS cells as judged by measuring the lengths of CldU tracts (Fig. 4b, c) and also by plotting replication fork speeds (Supplementary Fig. 4a) that were calculated based on the assumption that 1 μm of fiber corresponds to 2.59 kb[34]. As replication fork slowing and shorter tracts can, in this assay, result from two scenarios, i.e., either a slower overall DNA polymerization rate or increased frequency of fork stalling[35], we analyzed the fates of two (sister) replication forks emanating in opposite directions from the same origin. To this end, we calculated the ratio of the lengths of CldU tracts of sister replication forks (length of the shorter tract divided by the length of longer tract); in uninfected cells, the ratio was ~1, indicating that sister forks traveled at similar speed. In *H. pylori*-infected AGS cells, the ratio dropped to ~0.6, indicating fork asymmetry and selective slowing/stalling of one fork only (Supplementary Fig. 4b). Replication fork slowing was not only observed upon *H. pylori* infection, but also upon treatment of AGS cells with the β- but not the α-anomer of ADP-heptose (Fig. 4b, c, Supplementary Fig. 4a). AGS cells lacking ALPK1 or TIFA expression were resistant to replication fork slowing in this assay, both in the setting of live infection and of exposure to β-ADP-heptose (Fig. 4b, c, Supplementary Fig. 4a). Complementation of *TIFA*-deficient AGS cells by TIFA overexpression from the lentiviral MND promoter rescued the effects of live *H. pylori* infection on fiber shortening (Supplementary Fig. 4c). The effect size of *H. pylori* infection on replication fork progression was comparable to the effects of the topoisomerase inhibitor camptothecin[36], and to the effects of the G quadruplex DNA (G4) ligand pyridostatin, a well-known inducer of both DNA damage and genome instability[37], which served as our positive controls (Fig. 4d). The slowing of replication forks upon *H. pylori* infection was dependent on RfaE and the Cag-PAI, as both null mutants failed to cause CldU tract shortening (Fig. 4e, f). TNF-α exposure did not cause CldU tract shortening (Fig. 4e, f), and also failed to rescue the phenotype of the RfaE mutant (Supplementary Fig. 4d). Exposure of infected cells to the NF-κB inhibitor BAY 11-7082 or to the transcription inhibitor triptolide rescued *H. pylori*-induced fork slowing (Fig. 4e, f). In contrast, treatment with the antioxidant *N*-acetyl-cysteine did not prevent fork slowing (Supplementary Fig. 4d). The combined results suggest that active transcription driven by NF-κB as a consequence of β-ADP-heptose delivery and ALPK1/TIFA signaling is a prerequisite of replication fork slowing in *H. pylori*-infected cells; in contrast, ROS produced by infected cells do not contribute to DNA damage or replication stress.

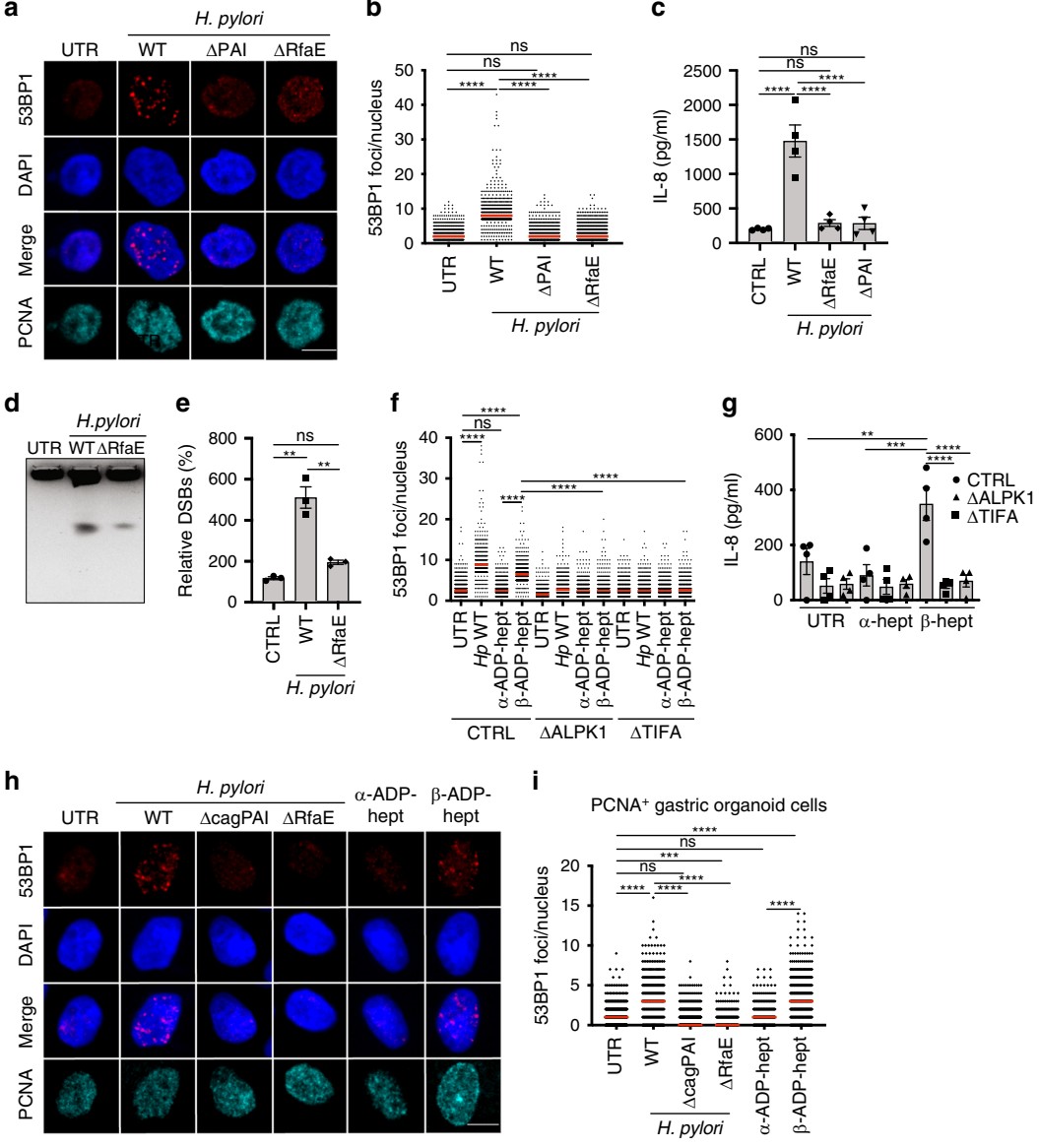

**Fig. 3 The bifunctional RfaE enzyme is required, and its product β-ADP-heptose is sufficient for the induction of DNA DSBs. a–c** AGS cells were infected for 6 h with *H. pylori* P12 or its isogenic RfaE and Cag-PAI mutants at an MOI of 50 and subjected to immunofluorescence staining for 53BP1 and PCNA as well as DAPI. Representative images are shown in **a** alongside scatter dot plots of >730 and up to 1655 cells per condition in **b**. IL-8 secretion as assessed by ELISA is shown in **c**. Data in **b** and **c** are pooled from four independent experiments. Red lines indicate medians. Data in **c** are represented as mean $+/-$ SEM of four independent experiments. Scale bar in **a**, 10 µm. **d, e** AGS cells were infected for 6 h as described above and subjected to PFGE. A representative gel is shown in **d** alongside the quantification of three gels (independent experiments) in **e**. Fragmented DNA was normalized to the intact DNA retained in the slot, and to the uninfected control condition, which was set as 100%. Data are presented as mean values $+/-$ SEM. **f, g** Wild-type (CTRL), ALPK1-deficient (ΔALPK1), and TIFA-deficient (ΔTIFA) AGS cells were exposed to α- or β-ADP-heptose at 0.5 µM final concentration for 6 h and subjected to immunofluorescence staining for 53BP1 and PCNA as well as DAPI. Scatter dot plots of >579 and up to 706 cells per condition are shown in **f**, with red lines indicating medians. IL-8 secretion as assessed by ELISA is shown in **g**. Data in **f** and **g** are pooled from four independent experiments. *P*-values were calculated by one-way ANOVA with Dunn's multiple comparisons correction. Data in **g** are presented as mean values $+/-$ SEM. Scale bar, 10 µm. **h, i** Gastric organoids were transferred to 2D cultures and infected with the indicated strains of *H. pylori* P12 (MOI of 50) or exposed to α- or β-ADP-heptose at 0.5 µM final concentration for 6 h. Cells were subjected to immunofluorescence staining for 53BP1 and PCNA as well as DAPI. Representative images are shown in **h** of 53BP1 foci (scale bar, 10 µm), alongside scatter dot plots of >298 and up to 603 cells per condition for PCNA⁺ cells (in **i**). Data in **i** are pooled from two independent experiments with cells derived from two different donors. Red lines indicate medians. *P*-values were calculated by one-way ANOVA with Dunn's multiple comparisons correction. ns, not significant; \*\**p* < 0.01, \*\*\**p* < 0.005, \*\*\*\**p* < 0.0001.

**R-loop formation is required for replication stress.** Active replication and transcription that co-occur in the same regions of the genome can result in replication stress and DNA damage if both machineries collide. Replication fork stalling at sites of these conflicts is caused by the formation of co-transcriptional R-loops[38–40]. To address whether R-loop formation is the cause of

DNA damage and replication stress induced by *H. pylori*, we used a cell line that conditionally expresses human RNase H1[41], an enzyme that cleaves the RNA strand in RNA/DNA hybrids and thereby eliminates R-loops. This cell line is derived from U2OS osteosarcoma cells that have previously been shown to be susceptible to *H. pylori*-induced DNA damage[15,18] and that are

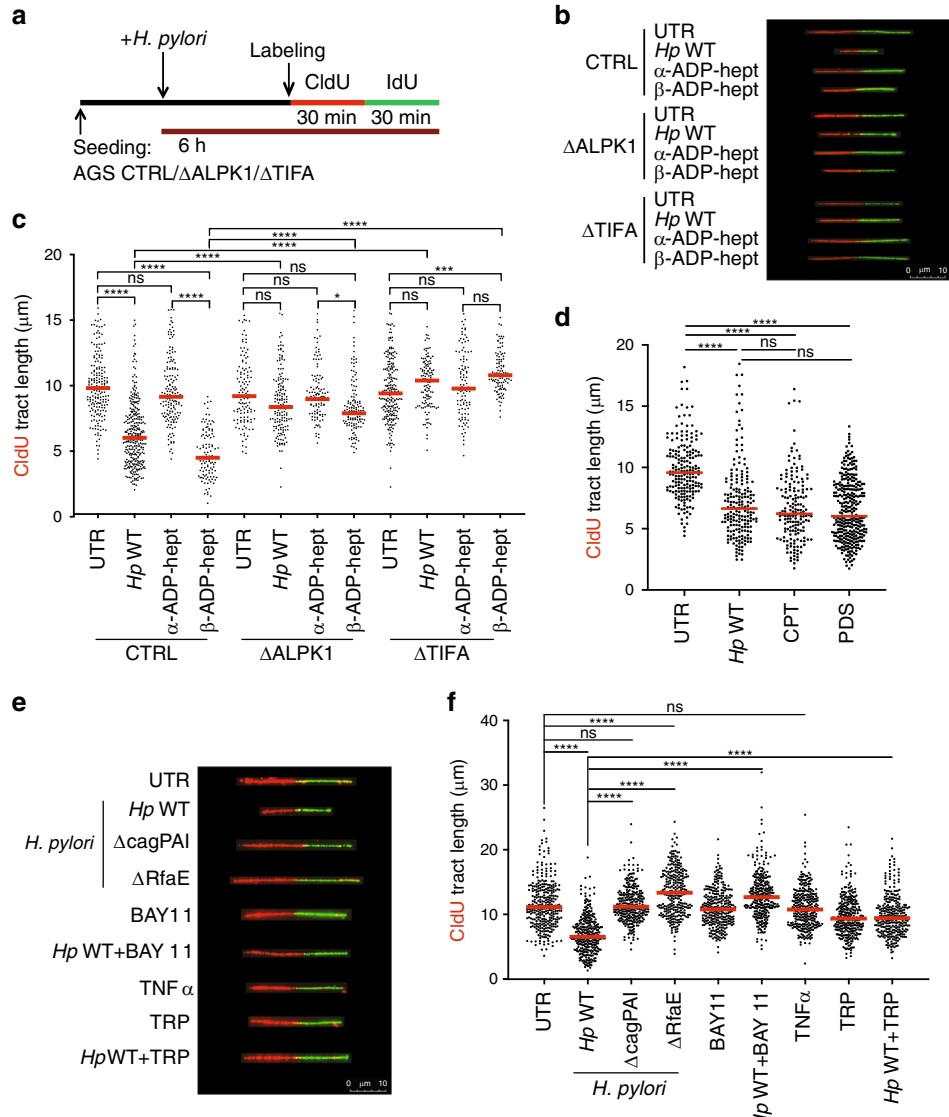

**Fig. 4 H. pylori induces DNA replication stress in host cells. a** Schematic representation of the protocol used for the quantification of the length of newly synthesized DNA tracts (fibers). **b, c** Wild-type (CTRL), ALPK1-deficient (ΔALPK1), and TIFA-deficient (ΔTIFA) AGS cells were either infected for 6 h with H. pylori strain P12 (MOI of 50) or treated with 0.5 μM α- or β-ADP-heptose and then labeled sequentially with CldU and IdU as shown in **a**. Representative DNA fibers are shown in **b** and scatter dot plots of CldU tract length (in μm) are shown in **c** for the indicated conditions. At least 106 and up to 318 fibers were analyzed per condition. Data in **c** are pooled from two independent experiments. **d** Wild-type AGS cells were either infected for 6 h with H. pylori strain P12 (MOI of 50) or treated with 100 nM camptothecin (CPT) or 10 μM pyridostatin (PDS) for 6 h and then labeled sequentially with CldU and IdU as shown in **a**. At least 200 fibers were analyzed per condition. Data in d are pooled from two independent experiments. **e, f** AGS cells were infected for 6 h with either the wild-type H. pylori strain P12 or its isogenic RfaE and Cag-PAI mutants (MOI of 50) and/or treated with the NF-κB inhibitor BAY 11-7082 (1 μM), the transcription inhibitor triptolide (100 nM), or TNF-α (10 nM) and labeled sequentially with CldU and IdU during the last 60 min of infection as shown in **a**. Representative DNA fibers are shown in **e** and scatter dot plots of CldU tract length (in μm) are shown in **f** for the indicated conditions. At least 141 and up to 213 fibers were analyzed per condition. Data in **f** are pooled from two independent experiments. Red lines indicate medians throughout. P-values were calculated throughout by one-way ANOVA with Dunn's multiple comparisons correction. ns, not significant; *$p < 0.05$, ****$p < 0.0001$.

commonly used in DNA damage research. Exposure of U2OS cells to wild-type H. pylori for six hours induced the formation of 53BP1 foci that increased with the MOI and were specific to S-phase, and were comparable in extent to those observed in AGS cells (Fig. 5a, b, left panels −DOX, Supplementary Fig. 5a, b). This DNA damage phenotype was accompanied by replication fork slowing as determined by DNA fiber assay (Fig. 5c, d, Supplementary Fig. 5c). As observed for AGS cells, replication fork slowing in H. pylori-infected U2OS cells was dependent on a functional Cag-PAI and RfaE, and could be completely reversed

by blocking transcription with the inhibitor triptolide or the NF-κB inhibitor BAY 11-7082 (Supplementary Fig. 5d, e). Both DNA damage in S-phase cells and replication stress could be induced also in U2OS cells by the addition of β- but not α-ADP-heptose (Fig. 5e–h, upper and left panels, −DOX, Supplementary Fig. 5f). Importantly, the induction of RNase H1 expression by doxycycline abrogated both 53BP1 foci formation (Fig. 5a, b, right panels, +DOX) and replication fork slowing (Fig. 5c, d, lower and right panels, +DOX, Supplementary Fig. 5c) upon H. pylori infection, and also upon β-ADP-heptose treatment (Fig. 5e–h).

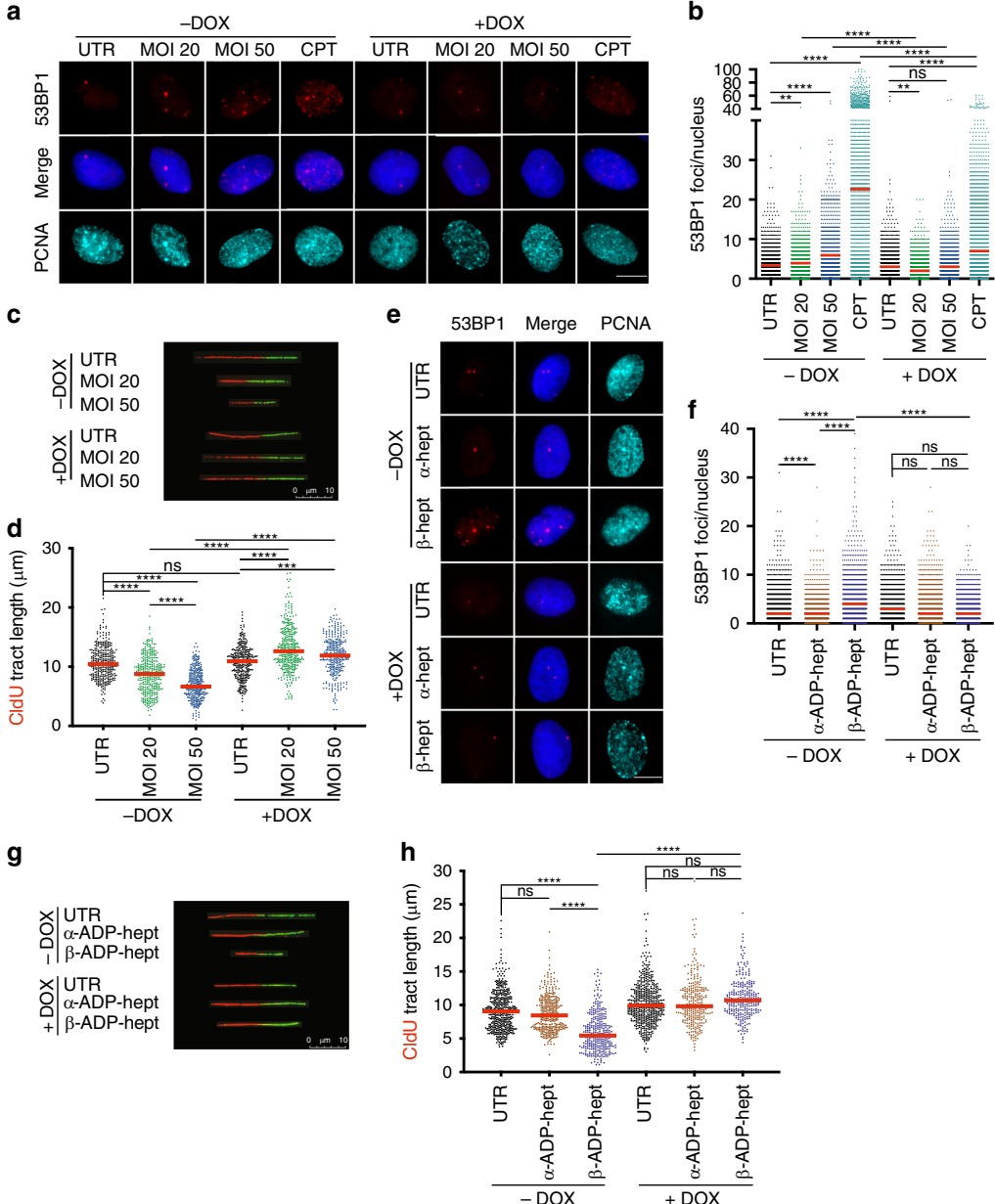

**Fig. 5 _H. pylori_-induced DNA damage and replication stress is prevented by overexpression of RNase H1. a, b** U2OS cells were either infected for 6 h with _H. pylori_ P12 (MOI of 20 or 50), or treated with 100 nM camptothecin (CPT), and were treated or not with doxycycline (−/+ DOX) to induce the expression of RNase H1. Cells were subjected to immunofluorescence staining for 53BP1 and PCNA as well as DAPI. Representative images are shown in **a** alongside scatter dot plots of >1382 and up to 1752 cells per condition in **b**. Data in **b** are pooled from three independent experiments. **c, d** U2OS cells were infected as described in **a, b** and additionally labeled sequentially with CIdU and IdU as shown in Fig. 4a for the assessment of replication tract length. Representative DNA fibers are shown in **c** and scatter dot plots of CIdU tract length (in μm) are shown in **d** for the indicated conditions. At least 100 and up to 500 fibers were analyzed per condition. Data in **d** are pooled from three independent experiments. **e, f** U2OS cells were exposed to α- or β-ADP-heptose at 0.5 μM final concentration for 6 h and treated or not with doxycycline (−/+ DOX) to induce the expression of RNase H1. Cells were subjected to immunofluorescence staining for 53BP1 and PCNA as well as DAPI. Representative images are shown in **e** alongside scatter dot plots of >1500 and up to 1752 cells per condition in **f**. Data in **f** are pooled from three independent experiments. Scale bar in **a** and **e**, 10 μm. **g, h** U2OS cells were exposed to α- or β-ADP-heptose at 0.5 μM final concentration for 6 h, treated or not with doxycycline (−/+ DOX) to induce the expression of RNase H1, and additionally labeled sequentially with CIdU and IdU for the assessment of DNA fiber tract length. Representative DNA fibers are shown in **g** and scatter dot plots of CIdU tract length (in μm) are shown in **h** for the indicated conditions. At least 299 and up to 514 fibers were analyzed per condition. Data in **h** are pooled from three independent experiments. Red lines indicate medians throughout. In **b, d, f,** and **h,** −DOX and +DOX samples of the same condition are color-coded to facilitate comparisons. _P_-values were calculated by one-way ANOVA with Dunn's multiple comparisons correction. ns, not significant; **$p < 0.01$, ***$p < 0.005$, ****$p < 0.0001$.

The analysis of sister forks emanating from the same origin showed that not only the slowing of fork progression, but also the fork asymmetry associated with *H. pylori* infection, could be reversed by induction of RNase H1 (Supplementary Fig. 5g, h). The combined results implicate R-loops, processed by RNase H1, in DNA damage and replication stress induced by *H. pylori*.

**R-loops form in a Cag-PAI-, RfaE-, and NF-κB-dependent manner**. The dependence of DNA damage and replication stress on R-loops prompted us to examine these structures in more detail, and to address the prerequisites of their formation. To this end, we used a cell line in the U2OS background that inducibly expresses a GFP-tagged RNase H1 harboring a point mutation (D210N) in its nuclease active site (RNase H1$^{D210N}$/GFP)[41], rendering it enzymatically inactive. Such inactive RNase H1 molecules bind their target structures but fail to resolve them; RNase H1$^{D210N}$/GFP foci forming in this cell line thus report the presence of R-loops. The addition of doxycycline to these reporter cells and concomitant *H. pylori* infection led to the appearance of on average 10–20 (and up to 80) RNase H1$^{D210N}$/GFP foci per cell that were not seen without doxycycline and that were clearly dependent on the MOI (Fig. 6a, b, Supplementary Fig. 6a). The extent of R-loop formation upon infection was comparable to that induced by the topoisomerase inhibitor camptothecin (Fig. 6a, b), a known inducer of R-loops[19]. RNase H1$^{D210N}$/GFP foci did not co-localize with sites of DNA damage (i.e., 53BP1 foci; Fig. 6a). Treatment of the reporter cells with β-ADP-heptose, but not the α-anomer, also resulted in R-loop formation (Fig. 6c, d) with a similar effect size as observed for live *H. pylori* infection or camptothecin treatment. Importantly, induction of expression of the catalytically inactive mutant of RNase H1 failed to resolve *H. pylori*- or β-ADP-heptose-induced DNA damage as assessed by the quantification of 53BP1 foci (Fig. 6a, c, e), indicating that the reduced DNA damage observed in settings of wild-type RNase H1 overexpression indeed results from its enzymatic activity. R-loop formation upon *H. pylori* infection was also abrogated by inhibition of transcription with triptolide, and by inhibition of NF-κB activation (Fig. 6f, g); the same treatments also blocked the formation of DNA damage in U2OS cells (Fig. 6f, h), as shown above for AGS cells (Fig. 2). The *rfaE* and *Cag-PAI* mutants of *H. pylori* failed to induce RNase H1$^{D210N}$/GFP foci (Fig. 6f, g). Infection-induced RNase H1$^{D210N}$/GFP foci were highly specific for PCNA$^+$ S-phase cells and were not observed in PCNA$^-$ cells (Fig. 6a–h, Supplementary Fig. 6b, c). Interestingly, the induction of RNase H1$^{D210N}$/GFP foci and of DNA damage correlated strongly with the level of incorporation of 5-fluorouracil (Supplementary Fig. 6d), which was also strongly dependent on *rfaE*, the Cag-PAI and NF-κB, and which we used as an indicator of RNA synthesis. Staining of infected cells with an antibody specifically recognizing RNA/DNA hybrids (clone S9.6) confirmed that infection with wild-type *H. pylori*, but not its *rfaE* and *Cag-PAI* mutants, induces R-loops not only in U2OS cells as shown with the reporter cell line, but also in AGS cells (Supplementary Fig. 6e, f). Taken together, the results suggest that the active transcription (of NF-κB target genes) that is induced by *H. pylori* in an β-ADP-heptose/ALPK1/TIFA-dependent manner leads to R-loop formation in actively replicating cells, possibly at sites where the transcription and replication machineries collide.

**H. pylori-infected cells show evidence of genotoxicity**. Our experimental data from two human cell lines as well as primary gastric cells are consistent with the hypothesis that transcription/replication conflicts generate substantial DNA damage during *H. pylori* infection. Such widespread DNA damage, especially if inadequately repaired, results in the subsequent accumulation of

mutations and an associated high risk of malignant transformation. As *H. pylori* is tightly associated with gastric carcinogenesis both epidemiologically and in various animal models, we investigated possible links between DNA damage, mutational burden, and gastric carcinogenesis. We first examined whether we would find evidence for *H. pylori*-associated genotoxicity in AGS cells by using the well-established cytokinesis-block micronucleus assay, a simple and sensitive procedure for reading out chromosome damage. AGS cells were treated, either already before or only during infection, with cytochalasin B, an inhibitor of the mitotic spindle that prevents cytokinesis. Cells that had completed one division were identified by their binucleated appearance, and the presence of small DAPI-stained fragments (micronuclei) that are indicative of chromosome breaks or intact mis-segregated chromosomes was quantified. We found up to three micronuclei per *H. pylori*-infected AGS cell, and substantially more infected cells exhibited evidence of micronuclei formation than control cells (Fig. 7a, b). The RfaE mutant failed to induce micronuclei (Fig. 7a, b), suggesting that *H. pylori* induces genotoxicity in an RfaE-dependent manner. The extent of genotoxicity induced by *H. pylori* would be expected to result in checkpoint activation, a DNA damage response and apoptosis. However, infected cells did not undergo apoptosis (Fig. 7c, d) and checkpoint activation— readout as phosphorylation of the DNA damage sensor ATM and its target KAP1— is very weak upon *H. pylori* infection relative to other DNA damaging agents such as camptothecin (Supplementary Fig. 7a, b, Supplementary Data 8–12). We conclude from these data that the DNA damage induced by *H. pylori*, despite causing chromosome breaks and mis-segregated chromosomes, fails to induce a DNA damage response or apoptosis.

**Higher mutational burden in *H. pylori*-associated gastric cancer**. We next asked whether gastric cancer genomic data would possibly provide circumstantial evidence for the presence of *H. pylori* as a driving force of malignant transformation. To this end, we used the non-human transcript and genomic information that is present in publicly accessible multi-omics (RNA-sequencing, whole-genome sequencing, and whole-exome sequencing) datasets of gastric cancer, which had been generated as part of the Cancer Genome Atlas (TCGA) project, to annotate gastric cancer samples with their *H. pylori* status. We had access to multi-omics datasets for 291 tumors from treatment-naive gastric cancer patients that had previously been subjected to array-based somatic copy number analysis, whole-exome sequencing, and array-based DNA methylation profiling, and that had resulted in the description of four major genetically defined subtypes of gastric cancer[42]. These subtypes were EBV-positive gastric cancer (characterized by the presence of EBV genes and transcripts, a very low mutational burden and *PIK3CA* mutations), microsatellite-instable gastric cancer (high mutational burden, MSI high), chromosomally instable gastric cancer (with large numbers of copy number variations and a high mutational burden, and ubiquitous TP53 mutations) and genomically stable gastric cancer (low mutational burden, diffuse type by histology, early-onset, *RHOA* and *ARHGAP6/26* somatic genomic alterations)[42]. We were able to recapitulate the stratification of the 291 patients into these subtypes (Fig. 7e). We found evidence of *H. pylori* presence in all four subtypes, with 1/3 to 1/2 of patients of each subtype exhibiting evidence of *H. pylori* transcripts or genomic DNA in either their tumor and/or adjacent tissue (Fig. 7e). When comparing the tumor mutational burden of the four subtypes, we found that tumors from patients with evidence of *H. pylori* infection (in either tumor and/or adjacent tissue) of the most common MSI and CIN subtypes had a higher mutational burden than those without *H. pylori*

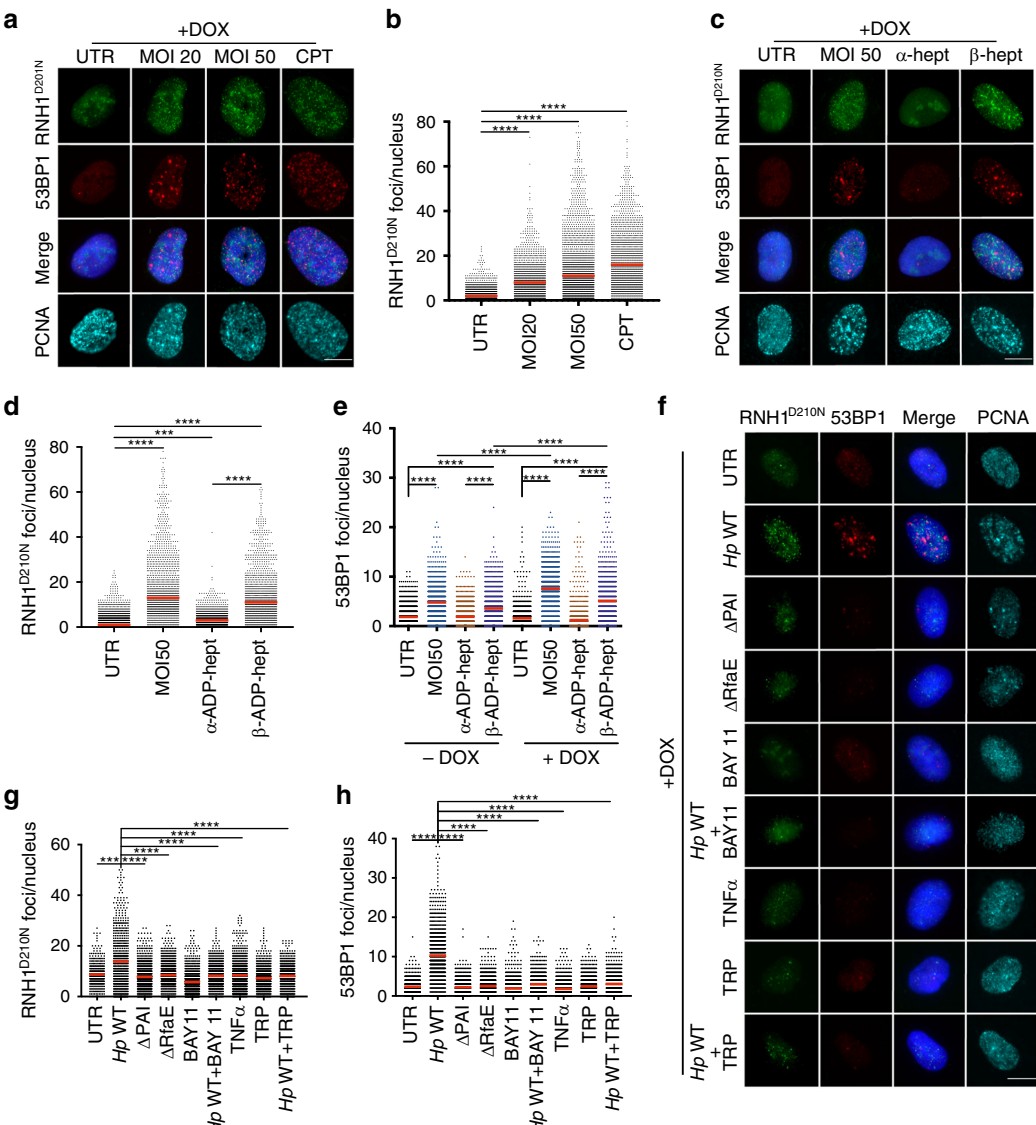

**Fig. 6 _H. pylori_ induces R-loop formation that depends on NF-κB and active transcription. a, b** U2OS cells were either infected for 6 h with _H. pylori_ P12 (MOI of 20 or 50), or treated with 100 nM camptothecin (CPT), and were treated or not with doxycycline (−/+ DOX) to induce the expression of a (D210N) mutant version of RNase H1 fused to GFP (RNH1^{D210N}). Cells were subjected to immunofluorescence staining for 53BP1 and PCNA as well as DAPI. Representative images are shown in RNH1^{D210N}/GFP foci and 53BP1 foci, alongside scatter dot plots of RNH1^{D210N}/GFP foci of >1468 and up to 1661 cells per condition in **b**; the −DOX panel shows lack of background signal in the GFP channel in the absence of RNH1^{D210N}/GFP expression. Data in **b** are pooled from three independent experiments. **c, d** U2OS cells were infected with _H. pylori_ (MOI of 50) or exposed to α- or β-ADP-heptose at 0.5 μM final concentration for 6 h and treated or not with doxycycline (−/+ DOX) to induce the expression of RNH1^{D210N}/GFP. Cells were subjected to immunofluorescence staining for 53BP1 and PCNA as well as DAPI. Representative images are shown in **c** of RNH1^{D210N}/GFP foci and 53BP1 foci, alongside scatter dot plots of RNH1^{D210N}/GFP foci of >1326 and up to 1565 cells per condition in **d**. Data in **d** are pooled from three independent experiments. **e** Scatter dot plots showing 53BP1 foci of >1000 and up to 3000 cells per condition of the U2OS cells shown in **a** and **c**, and their counterparts not treated with doxycycline. **f–h** U2OS cells were infected with the indicated strains of _H. pylori_ (MOI of 50) and/or exposed to the NF-κB inhibitor BAY 11-7082 (1 μM) or triptolide (100 nM) and treated with doxycycline to induce the expression of RNH1^{D210N}/GFP. Cells were subjected to immunofluorescence staining for 53BP1 and PCNA as well as DAPI. Representative images are shown in **f** of RNH1^{D210N}/GFP foci and 53BP1 foci, alongside scatter dot plots of RNH1^{D210N}/GFP foci of >405 and up to 868 cells per condition in **g**, and of 53BP1 foci of >691 and up to 1042 cells per condition in **h**. Data in **g** and **h** are pooled from three independent experiments. Red lines indicate medians throughout. _P_-values were calculated by one-way ANOVA with Dunn's multiple comparisons correction. ns, not significant; ***_p_ < 0.005, ****_p_ < 0.0001. Scale bars, 10 μm.

infection: within the CIN subtype, which with 136 samples was the most common of the four gastric cancer subtypes in the TCGA dataset, 57 _H. pylori_-infected patients had on average 2 +/− 1.5 mutations per megabase, whereas this number was 1.54 +/− 0.98 in the 79 _H. pylori_-negative patients (_p_ = 0.052 as determined by student's _T_-test). Within the MSI subtype, the figures were 20 +/− 16 for 23 _H. pylori_-positive vs. 16 +/− 7 for

33 _H. pylori_-negative patients, but the difference was not statistically significant (_p_ = 0.2). A total of 232 genes were differentially affected by mutations as a function of _H. pylori_ infection across all 291 samples (Supplementary Data 13, Fig. 7e).

We also looked specifically for mutations and copy number variations (CNVs) in genes involved in R-loop prevention and processing, with the idea that mutations in such genes might

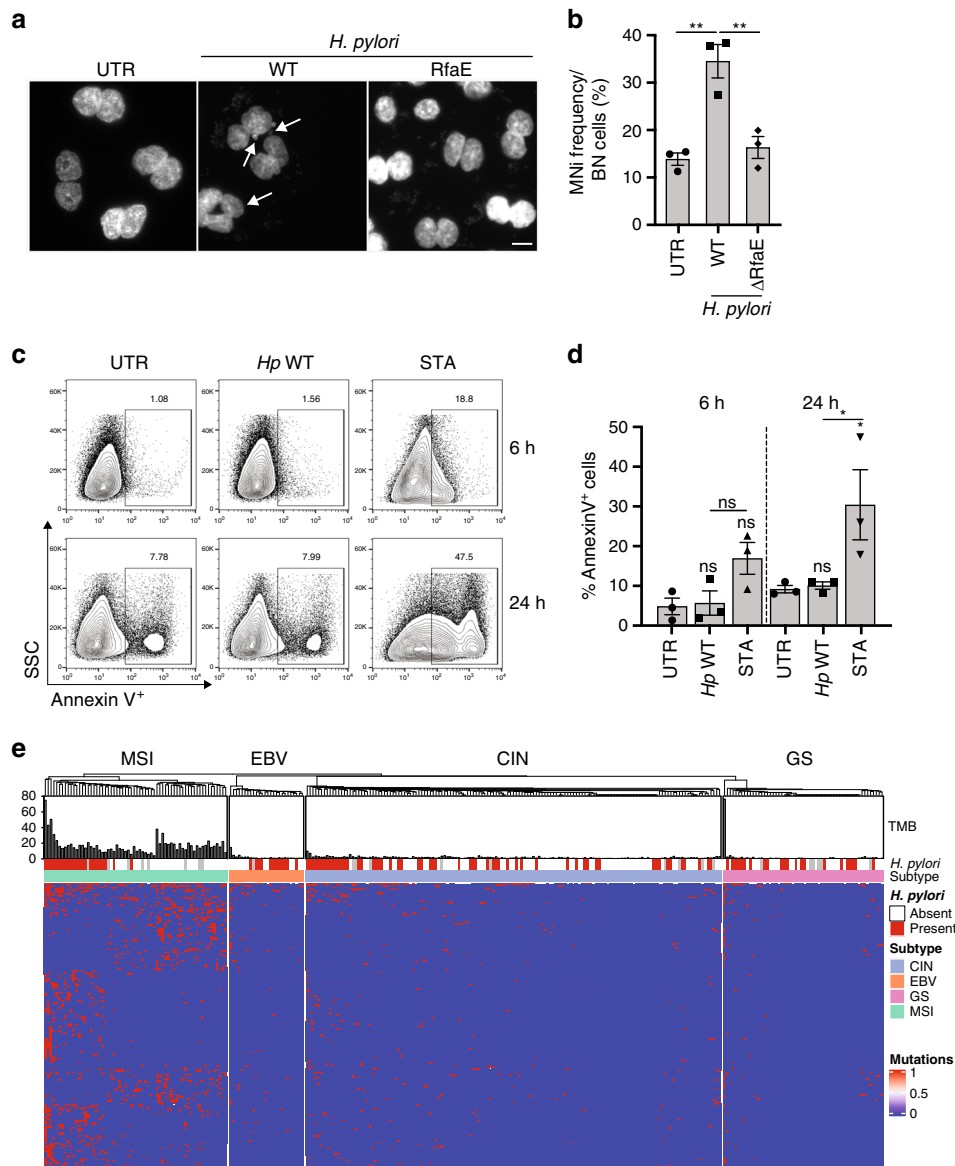

**Fig. 7 *H. pylori* induces micronuclei but not apoptosis in gastric epithelial cells and its presence is associated with specific mutational patterns in gastric cancer. a**, **b** Micronuclei formation as quantified in binucleated AGS cells that had been subjected to cytochalasin B treatment to prevent cytokinesis, and had additionally been infected for 16 h with the indicated WT or *RfaE* mutant strains of *H. pylori*. Representative images are shown in **a**; arrows point to micronuclei. Frequencies of binucleated (BN) cells with micronuclei are quantified in **b** and pooled from three independent experiments. Data are presented as mean values +/− SEM. *P*-values were calculated by one-way ANOVA with Dunn's multiple comparisons correction. Scale bar represents 15 μm. **c**, **d** Annexin V staining for apoptosis of AGS cells infected with WT *H. pylori*, or treated with 5 μm staurosporine (STA) for 6 or 24 h. Representative FACS plots are shown in **c**, and the quantification of Annexin V-positive cells of three independent experiments is shown in **d**. Data are presented as mean values +/− SEM. *P*-values were calculated by two-way ANOVA with Tukey's multiple comparisons correction. **e** Mutational signature of *H. pylori* in gastric cancer as determined on 291 gastric cancer samples available through the Cancer Genome Atlas (TCGA). The *H. pylori* status was annotated based on the presence of at least one unambiguous *H. pylori*-specific read detected in the transcriptome, whole genome or exome of either the tumor or adjacent tissue of a total of 267 among the 291 patients. The gastric cancer subtype was assigned based on the previous description[42] of CNVs, presence of EBV transcripts, methylation of signature genes and microsatellite instability; the color code on the right indicates the four subtypes chromosomally instable (CIN), microsatellite instable (MSI), EBV-positive (EBV), and genomically stable (GS). The heatmap shows the 232 genes found to be differentially mutated between *H. pylori*-positive and -negative cases (*p* < 0.05 as determined by Student's *t*-test); the majority of differentially mutated genes are found in the MSI subtype. Genes are listed in Supplementary Table 1. ns, not significant; *p < 0.05, **p < 0.01.

predispose to gastric carcinogenesis. We plotted the mutations and CNVs in 18 genes with functions in R-loop prevention or processing relative to seven known gastric cancer driver genes, i.e., *CDH1*, *APC*, *TP53*, *ARID1A*, *PIK3CA*, *KRAS*, and *ERBB2*, and relative to the mutational burden per megabase. The 18 genes with R-loop metabolism-related functions were selected based on literature searches: we focused on the RNA/DNA helicases *AQR*,

*SETX*, and *DHX9* involved in R-loop unwinding[19,43,44], the DNA helicase *PIF1* known to prevent the accumulation of R-loops at tRNA genes[45], the DNA topoisomerase *TOP3B*, which also prevents accumulation of R-loops[46], the endonuclease *RNASEH2* which, like RNase H1 cleaves the RNA strand in RNA/DNA hybrids[47], the splicing factor *SRSF1*, known to prevent R-loop formation[48], *THOC1* and *THOC2*, components of the

THO/TREX complex involved in mRNA export and R-loop prevention[49], the THO/TREX complex associated histone deacetylase *SIN3A*[50], and the six components of the TREX-2 complex that, with *BRCA2*, is also associated with mRNA export and R-loop prevention[51]. Several of these genes were recurrently (in up to 36% of gastric cancer cases) affected by CNVs and/or missense or frameshift mutations, with the most recurrently affected genes belonging to the TREX-2 complex (*ENY2*, *MCM3AP, SEM1, CETN3*) or having RNA/DNA helicase or endonuclease activity (*SETX, AQR, DHX9, RNASEH2*) (Supplementary Fig. 7c). Mutations in R-loop metabolism genes were largely predicted to be damaging mutations, and—not unexpectedly—were mostly detected in the microsatellite-instable subtype of gastric cancer that exhibited the highest overall mutational burden; CNVs affecting R-loop genes were mostly detected in the chromosomally instable subtype (Supplementary Fig. 7c). In conclusion, the mutational landscape of gastric cancer provides circumstantial evidence for the hypothesis that *H. pylori* infection favors chromosomal instability, and that the inactivation of genes encoding helicases, RNases and other factors involved in R-loop prevention or processing may be an early event in gastric carcinogenesis, predisposing *H. pylori*-infected cells to aberrant R-loop accumulation and its potentially deleterious consequences.

## Discussion

It is now widely accepted that Gram-negative bacteria are sensed by epithelial cells via their production of LPS biosynthetic intermediates, which bind directly to ALPK1 to induce downstream TIFA/NF-κB signaling. The mechanisms through which LPS intermediates such as HBP and β-ADP-heptose reach the host cell cytosol likely differ across bacterial species and, in the case of *H. pylori*, appear to involve a functional T4SS[21–23]. Our data corroborate this notion, as the loss of a key enzyme (RfaE) involved in HBP and β-ADP-heptose synthesis phenocopies the loss of the T4SS-encoding Cag-PAI of *H. pylori* in all DNA-damage-related settings examined here. Other functions of the Cag-PAI, such as the delivery of the only known T4SS protein substrate CagA, are unaffected by RfaE deficiency. We and others have reported previously that the DNA damage induced by *H. pylori* in its epithelial target cells is dependent on a functional T4SS[16–18], and is largely restricted to transcribed regions of the genome[17]. These earlier observations are consistent with the model emerging from the current findings, where co-occurring active replication and transcription in S-phase cells leads to R-loop formation, which is followed by the processing of both the RNA and the DNA strands of the RNA/DNA hybrid by dedicated enzymes (Supplementary Fig. 8). The nucleolytic processing of DNA in such structures is in turn sensed by DNA repair factors such as 53BP1 and can be readily visualized and quantified by pulsed-field gel electrophoresis and immunofluorescence microscopy.

NF-κB has emerged in the course of our studies as the dominant transcription factor driving the early, ALPK1/TIFA-dependent response to *H. pylori*. The increase in incorporation of 5-FU —which served as our readout of de novo RNA synthesis— due to *H. pylori* infection was almost completely blocked not only by a general inhibitor of RNA-Pol II, but also by a highly specific inhibitor of NF-κB nuclear translocation. The same treatments prevented R-loop formation, replication fork slowing, and DNA damage. NF-κB was initially established as a key molecular link between inflammation and cancer[52] by two seminal studies showing that NF-κB has tumor-promoting properties in settings of inflammation-associated cancers of the colon and the liver[53,54]. NF-κB promotes carcinogenesis in such particularly susceptible organs through both tumor-cell-intrinsic and -extrinsic mechanisms (recently reviewed in ref. [52]). NF-κB on

the one hand acts directly in tumor cells by promoting the production of reactive oxygen and nitrogen species, which induce DNA damage and oncogenic mutations[55] and by promoting cell survival through the activation of anti-apoptotic genes such as BCL-X$_L$ and BCL-2[56,57]. On the other hand, NF-κB activation in various cell types of the tumor microenvironment, especially myeloid cells and cancer-associated fibroblasts, promotes tumor growth and dissemination through the production of cytokines (with TNF-α and IL-6 being the best-understood pro-tumorigenic NF-κB targets), chemokines, and pro-angiogenic factors[58]. Our data additionally implicate NF-κB activation in genomic instability through co-transcriptional R-loop formation that is associated with replication stress and DNA damage. Not every pathway to NF-κB activation is equally prone to R-loop induction and DNA damage; we have extensively tested TNF-α, which activates IKK and NF-kB via the TRADD, TRAF2, NIK signaling axis[59] and bypasses ALPK1 and TIFA, and never found evidence of R-loops or DNA damage in this setting. TNF-α also fails to synergize with the non-genotoxic *RfaE* mutant to induce DNA damage. These results indicate that NF-κB signaling is required, but not sufficient, to induce DNA damage in actively transcribing/ replicating cells.

Two scenarios are conceivable that are not easily distinguished experimentally but would both explain R-loop formation in the context of *H. pylori*-induced NF-κB activation. As shown by multiple laboratories in various models systems—ranging from bacteria, to yeast, to mammalian cells—R-loops may either form as a consequence of head-on collisions between a replication fork and transcription bubble as proposed previously, or alternatively, may form in highly transcribed regions of the genome, where they cause pausing of RNA-Pol II, which in turn blocks the progression of oncoming replication forks[38,60–63]. Irrespective of the sequence of events, we find that R-loop accumulation upon *H. pylori* exposure promotes S-phase-specific DNA damage, as overexpression of RNase H1 reduces DNA damage and abrogates replication stress. Both suppressive effects of RNase H1 require its enzymatic activity as they are not observed with a catalytically dead version of the enzyme. R-loops have been linked not only to DNA damage, genomic instability, and chromosomal rearrangements, but also to oncogenesis[51,61,64–66]. Increased transcriptional activity resulting from oncogene activation is known to promote R-loop accumulation and replication stress. This has been well documented for HRAS$^{V12}$ overexpression, which through elevated expression of the general transcription factor TATA-box binding protein leads to increased RNA synthesis, R-loop accumulation, replication fork slowing, and DNA damage[66]. In breast cancer cells, R-loops accumulate and drive DNA damage in heavily transcribed estrogen-induced genes, and translocations and structural variants are found in genes induced by estrogen signaling[67]. In the scenarios described by these two studies, the transcriptional landscape of cancer cells appears to dictate where DNA damage occurs through R-loop formation. Future studies integrating data from ChIP-seq-based surveys (for example, by pull-down of R-loop-associated DNA with the RNA/ DNA hybrid-specific antibody S9.6) with transcriptional profiling and mutational data for gastric cancer and its precursor lesions will reveal whether certain heavily transcribed (NF-κB-regulated) loci are particularly prone to R-loop-dependent DNA damage and mutagenesis.

Several molecular mechanisms have been implicated in R-loop-induced DNA damage. On the one hand, persistent RNA/DNA hybrids may compromise DNA repair by blocking access of DNA repair factors to DNA DSBs[68], or by reducing the efficiency of DNA end resection, which determines whether repair proceeds via homologous recombination or non-homologous end joining[69]. An alternative mechanism that has been proposed to link R-loops to DNA damage posits that nucleotide excision repair

(NER) enzymes XPG and XPF cut the DNA in the RNA/DNA hybrid, producing a single-strand gap that is converted into a DSB by replication or additional strand breaks[19]. Having shown previously that the depletion of XPF or XPG reduces DNA DSBs upon *H. pylori* infection[18], our combined data are consistent with a model where the processing of R-loops by NER factors, rather than compromised DNA repair in the face of R-loops, link R-loop formation to DNA damage (Supplementary Fig. 8).

The physiological prevention and/or processing of R-loops requires a number of factors (RNA processing and splicing factors, helicases and nucleases), some of which were found in this study to be recurrently mutated or subject to copy number losses in gastric cancer. In particular, the genes encoding RNase H2 and the RNA/DNA helicase Aquarius (*AQR*), and genes encoding the components of the TREX-2 mRNA export complex—all known to be involved in R-loop processing or prevention[45,51,60,70] were recurrently subjected to CNVs or mutated in gastric cancer, and especially in the microsatellite instable and chromosomally instable subtypes of the disease. In agreement with the hypothesis that *H. pylori*-induced DNA damage causes genotoxicity and predisposes to mutagenesis, we find more mutations, and different mutation patterns, in gastric cancer samples of patients with evidence of *H. pylori* presence at the time of diagnosis. As the diagnosis of *H. pylori* presence in non-human sequences of gastric cancer and adjacent tissue is prone to false-negative (not so much false-positive) results, the differences observed in the mutational burden need to be interpreted with caution; this is especially true because *H. pylori* disappears from its gastric niche as intestinal metaplasia and other gastric cancer precursor lesions form. We propose that inactivating mutations or copy number losses in genes involved in the prevention or elimination of R-loops are early events in gastric cancer that predispose hyper-proliferating cells to *H. pylori*-induced R-loop accumulation and DNA damage. Immunohistochemical staining of consecutive sections for proliferating cells and for *H. pylori* suggests that live bacteria may indeed come in direct contact with replicating cells in gastric corpus glands of gastritis patients. In summary, we propose here a mechanism of transcription-associated R-loop formation linked to DNA damage that may connect innate immunity to DNA damage, and bacterial infection to carcinogenesis.

## Methods

**Cell culture, bacterial strains, and infection conditions**. Wild-type AGS cells (ATCC CRL 1739, a human gastric adenocarcinoma cell line) and AGS cells subjected to TIFA and ALPK1 deletion by CRISPR, and to TIFA complementation with the wild-type *TIFA* gene[21,22] were grown in RPMI supplemented with 10% fetal calf serum (FCS), 100 U/ml penicillin, and 100 μg/ml streptomycin. U2OS (ATCC HTB96, a human osteosarcoma cell line) T-Rex cell lines carrying pAIO vectors for the expression of RNase H1/GFP or RNase H1D210N/GFP were cultivated in Dulbecco's modified Eagle's medium (DMEM) supplemented with 10% FCS (Tet-free approved), 100 U/ml penicillin, and 100 μg/ml streptomycin, 50 μg/ml hygromycin B and 1 μg/ml puromycin. Doxycycline (1 ng/ml) was added for 24 h to induce the expression of recombinant RNase H1 and to downregulate the endogenous RNH1 expression. BAY 11-7082 and triptolide were purchased from Sigma-Aldrich. *H. pylori* was grown on solid medium on horse blood agar containing 4% Columbia agar base (Oxoid; Basingstoke), 5% defibrinated horse blood (HemoStatLabs), 0.2% β-cyclodextrin, 5 μg/ml trimethoprim, 8 μg/ml amphotericin B, 10 μg/ml vancomycin, 5 μg/ml cefsulodin, and 2.5 U/ml poly-myxin B sulfate (all from Sigma-Aldrich) at 37 °C for 2 days under microaerophilic conditions. For liquid culture, *H. pylori* was grown in Brucella broth (Difco) containing 10% FBS (Life Technologies) with shaking in a microaerobic atmosphere at 37 °C. Bacterial numbers were determined by measuring the optical density at 600 nm, and bacteria were added to cells at an MOI of 20 or 50 for 6 h. The following strains of *H. pylori* were used: G27[71], P12 wild-type, P12ΔPAI[72], and P12ΔRfaE[22]. All *H. pylori* liquid cultures were routinely assessed by light microscopy for contamination, morphology, and motility prior to use in infections.

**Two-dimensional gastric organoid culture and *H. pylori* infection**. Human tissues were obtained from patients of the University Clinic, Wuerzburg. This study

was approved by the ethical committee of the University of Wuerzburg's University Clinic (approval 37/16). Gastric organoids were initiated from tissues[30] and for infection in 2D, were sheared to single cells and seeded in coated μ-slide coverslips (IBIDI, 8 Well, Cat. No. 80826) in antibiotic-free culture medium: Advanced Dulbecco's modified Eagle medium (DMEM)/F12 supplemented with 10 mmol/L HEPES, GlutaMAX, 1× B27, 1 mmol/L *N*-acetylcysteine, 50 ng/mL epidermal growth factor (EGF) (all from Invitrogen, Waltham, MA), 10% noggin-conditioned medium, 10% R-spondin1-conditioned medium, 50% Wnt-conditioned medium, 200 ng/mL fibroblast growth factor (FGF)10 (Peprotech, Hamburg, Germany), 1 nmol/L gastrin (Tocris, Bristol, UK), and 2 mmol/L transforming growth factor (TGF)βi (A-83-01; Tocris). After seeding, 10 mmol/L rho-associated coiled-coil forming protein serine/threoninekinase (RHOK) (Y-27632; Sigma-Aldrich) was added. The medium was refreshed every 2–3 days. Liquid cultures of *H. pylori* were prepared as described above. Cells were infected on day 7 or 8 at 50% confluency with an MOI of 50 or treated with α-ADP-heptose and β-ADP-heptose, respectively, at a final concentration of 0.5 μM for 6 h. Supernatants were collected for ELISA. Cells were subjected to immunofluorescence microscopy.

**Cytokinesis-block micronuclei assay**. Cells were grown on autoclaved coverslips and supplemented with 2 μg/ml Cytochalasin B prior to infection with *Helicobacter pylori* (P12 wild-type, P12ΔRfaE) for 12 h. Cells were washed three times with 1× PBS and then fixed with 4% paraformaldehyde for 15 min at RT. After washing three times with 1× PBS cells were stained with 1 μg/ml DAPI diluted in distilled water for 5 min at RT. Coverslips were mounted with Fluoromount-G mounting medium (Invitrogen, 00-4958-02). Images were acquired with a Leica DM6B fluorescent microscope. In each experiment micronuclei of at least 100 binucleated cells were counted per condition.

**Flow cytometric quantification of apoptosis and reactive oxygen species formation**. For quantification of reactive oxygen species, AGS cells were stained with 25 μM 2′,7′-dichlorofluorescin diacetate (DCFDA, Abcam, ab113851) and infected with *Helicobacter pylori* or treated with 50 μM Tert-Butyl Hydrogen Peroxide (TBHP, Abcam, ab113851) for 6 h. Cells were harvested, resuspended in 1× PBS/1%BSA and subjected to flow cytometry. Apoptosis was quantified by Annexin V staining. Cells were infected with *Helicobacter pylori* or treated with 5 μM Staurosporin for 6 or 24 h. After harvesting cells were stained with PE-Cy7-labeled Annexin V (BioLegend, Cat No. 559934) diluted in 1x Annexin V Binding buffer (BD Bioscience, Cat No 640950) for 15 min at RT in the dark. Data were acquired on a CyAn ADP (Beckman Coulter) flow cytometer and analyzed with the FlowJo software (TreeStar).

**5-Fluorouridine incorporation and staining**. Cells grown on coverslips were infected with *Helicobacter pylori* or treated as described above and pulse-labeled with 1 mM 5-Fluorouridine (F5130, Sigma-Aldrich) for the last 30 min of the experiment. After incubation, cells were washed with PBS, pre-extracted with pre-extraction buffer (25 mM Hepes-NaOH, pH 7.7, 50 mM NaCl, 1 mM EDTA, 3 mM MgCl2, 0.3 M sucrose, 0.5% Triton X-100) for 5 min on ice and fixed with 4% formaldehyde (F8775, Sigma-Aldrich) in PBS for 15 min (in the dark, room temperature), washed with PBS and fixed with ice-cold methanol for 20 min (in the dark, −20 °C). After fixation, coverslips were washed with PBS and blocked with 1% BSA in PBS for 10 min and then stained with mouse monoclonal anti-BrdU antibody (clone BU-33, which recognizes 5-Fluorouridine, B2531, Sigma-Aldrich) in 1% BSA in PBS and with polyclonal rabbit anti-PCNA antibody (ab18197, Abcam) for 120 min. After washing with PBS, coverslips were incubated with Alexa Fluor 568-conjugated secondary antibody (Invitrogen, a11031), AF647 goat anti-rabbit (Invitrogen, a21245) in 1% BSA in PBS, counterstained with DAPI and mounted using Fluoromount-G mounting media (Invitrogen). Images were captured on Olympus IX83 microscope equipped with ScanR imaging platform using 40×/0.9 NA dry objective. For analysis images were submitted to ScanR Analysis software based on signal intensity for FU. Nuclei were identified based on the DAPI signal. Approximately 1000 cells were measured per condition.

**Immunofluorescence microscopy**. Cells grown on autoclaved coverslips were infected with *Helicobacter pylori* or treated as described above for 6 h. After the infection/treatment, cells were washed three times with 1× PBS and then permeabilized for 5 min with pre-extraction solution (25 mM Hepes, pH 7.7; 50 mM NaCl; 1 mM EDTA; 3 mM MgCl2; 300 mM sucrose, 0.5% Triton X-100) on ice. After a brief wash with 1× PBS, cells were fixed with 4% paraformaldehyde for 15 min at RT in the dark. For PCNA immunofluorescence, after fixation with paraformaldehyde, cells were washed three times with 1× PBS and then fixed with ice-cold methanol for 20 min at −20 °C. After fixation, cells were washed three times with 1× PBS and blocked in 1% BSA/1× PBS for 20 min at RT. Coverslips were then incubated with appropriate primary antibodies diluted in 1% BSA/1× PBS for 90 min at RT in dark. The following antibodies and dilutions were used: anti-phospho Histone H2A.X (Ser139) mouse monoclonal (Millipore, 05-636-AF647, 1:300), anti-53BP1 rabbit polyclonal (Santa Cruz, sc-33760, 1:300), and anti-PCNA Mouse monoclonal (Santa Cruz, sc56, 1:250). Coverslips were washed three times with 1× PBS and then incubated with secondary antibodies diluted in 1% BSA/1× PBS for 30 min at RT in dark. Secondary antibodies and dilutions were

Alexa Fluor 488 Goat Anti-Rabbit IgG (Life Technologies, A11034, 1:400), Alexa Fluor 568 Goat Anti-Rabbit (Life Technologies, A11036, 1:400), Alexa Fluor 568 Goat Anti-Mouse (Life Technologies, A11031, 1:400), Alexa Fluor 647 Goat Anti-Rabbit (Life Technologies, A21245, 1:400), and Alexa Fluor 647 Goat Anti-Mouse (Invitrogen, A21235, 1:400). After three washes with 1× PBS, coverslips were stained with 1 µg/ml DAPI diluted in distilled water for 2 min at RT in dark. Coverslips were mounted with Fluoromount-G mounting medium (Invitrogen, 00-4958-02). Images were acquired with a Leica DM6B fluorescent microscope or Leica SP8 inverted confocal microscope. For the analysis of phosphorylated H2AX signal and 53BP1 foci, automated image acquisition was performed on a IX83 microscope (Olympus) equipped with ScanR imaging platform using a 40×/0.9 NA objective. The analysis of acquired images was performed using ScanR Analysis software. For the analysis of RNH1-GFP foci, images were acquired using GE IN Cell Analyzer 2500HS with a 40×/1.15 NA water-immersion objective and the analysis was performed using CellProfiler 3.1.5[73]. Nuclei were identified based on the DAPI signal and the parameters of interest were analyzed for each nuclear object using modules of the ScanR Analysis software or CellProfiler 3.1.5.

**Pulsed-field gel electrophoresis (PFGE).** For PFGE, cells were harvested, embedded in 1.5% agarose plugs ($5 \times 10^5$ cells/plug), digested in lysis buffer (100 mM EDTA, 1% (w/v) sodium lauryl sarcosyne, 0.2% (w/v) sodium deoxycholate, 1 mg/ml proteinase K) at 37 °C for 48 h and washed in TE (20 mM Tris-HCl (pH 8.0), 50 mM EDTA). Electrophoresis was performed for 23 h at 14 °C in 0.9% (w/v) pulsed-field-certified agarose (Bio-Rad) containing Tris-borate/EDTA buffer according to the conditions described in ref. [74] and adapted to a Bio-Rad CHEF DR III apparatus. Gels were stained with ethidium bromide and imaged on an Alpha Innotech Imager. Two images were taken of each gel with 8 and 100 ms exposure time, respectively. For quantification, the signal intensity of the fragmented DNA (which has migrated into the gel and formed a distinct band) after 100 ms exposure time was normalized to the signal intensity of intact DNA (retained in the loading pocket) of the same sample at 8 ms exposure.

**DNA fiber assay.** For the analysis of DNA fiber length, cells were sequentially pulse-labeled with 30 mM CldU (Sigma-Aldrich) and 250 mM IdU (European Pharmacopoeia) for 30 min each. The cells were collected and resuspended in PBS at $2.5 \times 10^5$ cells per ml. The labeled cells were diluted 1:1 (v/v) with unlabeled cells, and 3 µl of cells were mixed with 7 µl of lysis buffer (200 mM Tris-HCl (pH 7.5), 50 mM EDTA, and 0.5% (w/v) SDS) on a glass slide. After 9 min, the slides were tilted to 15–45°, and the resulting DNA spreads were air-dried and fixed in methanol/acetic acid (3:1) solution overnight at 4°. The DNA fibers were denatured with 2.5 M HCl for 60 min, washed several times with PBS to neutralize the pH, and blocked with 0.1% Tween 20 in 2% BSA/PBS for 40 min. The newly replicated CldU and IdU tracks were labeled for 2.5 h in the dark, at room temperature, with anti-BrdU antibodies recognizing CldU (rat, Abcamab6326, 1:500) and IdU (mouse, BD 347580 B44, 1:100), followed by 1 h incubation with secondary antibodies at room temperature in the dark: anti-mouse Alexa Fluor 488 (1:300, A11001, Invitrogen) and anti-rat Cy3 (1:150, 712-166-153, Jackson ImmunoResearchLaboratories). Fibers were visualized on a Leica DMI 6000 inverted microscope using an HCX Plan APO DIC 63x oil objective (1.4–0.6 NA) and analyzed using Fiji[75]. At least 100 fibers were analyzed per replicate condition. Fork speed in kb/min was calculated by multiplying the measured length in µm with a conversion factor of 2.59 kb/µm and dividing by the duration of the labeling pulse[34].

**Immunohistochemistry.** Consecutive gastric formalin-fixed and paraffin-embedded sections from six patients presenting with H. pylori-associated gastritis and three patients with normal gastric mucosa were either stained with hematoxylin and eosin (H&E) or pretreated with CC1 buffer (Ventana Roche) for 16 min at 100 °C prior to staining with either Ki67 rabbit monoclonal antibody (clone 30-9, Ventana Roche) or with anti-H. pylori polyclonal rabbit antibody (B0471, Dako) for 30 min at 36 °C. Immunohistochemical staining and detection were performed using the BenchMark Ultra system (Ventana Roche) and OptiView DAB IHC Detection kit (Roche). Slides were scanned using the Hamamatsu Nanozoomer HAT scanner and analyzed using NDP.view version 2. The study was approved by the Cantonal Ethics Committee of Zurich (KEK).

**Western blotting.** AGS cells were infected with H. pylori or treated as described above for 6 h. Cells were harvested and lysed using 50 µl RIPA buffer (Sigma-Aldrich, Cat. No. R0278-50ML) supplemented with 1× complete protease inhibitor cocktail (Roche). Protein concentrations were determined using Bradford assay (Bio-Rad, Cat No. 5000002) and equal amounts were separated by SDS/PAGE (10% gel) followed by transfer onto nitrocellulose membranes (GE Healthcare Life Sciences, Cat No. 10600023). Membranes were incubated with antibodies against β-Actin (CST, 3700), p-KAP1 (S824) (Abcam, ab70369), and p-ATM (Abcam, ab81292).

**TCGA data analysis and annotation of H. pylori status.** TCGA stomach adenocarcinoma mutation and copy number data were downloaded with the R package TCGAbiolinks[76]. Copy number thresholds were taken from the TCGA

CNV pipeline (−0.3 for loss and 0.3 for gain). Molecular subtyping was performed based on a previous publication[42]. The Oncoprint-like plot was generated using the R package ComplexHeatmap[77]. Damaging mutations were defined by having a "deleterious" annotation from SIFT and "damaging" annotation from PolyPhen. To annotate stomach adenocarcinoma (STAD) samples from TCGA with H. pylori status, we acquired whole-genome (WGS), whole-exome (WXS), and transcriptome (RNA-seq) sequencing data from TCGA via the application programming interface (API) of the National Cancer Institute's (NCI) Genomic Data Commons (GDC). Raw sequencing data in bam format were screened for H. pylori and other microbiota using the PathSeq pipeline[78], which is available through the Broad Institute's Genome Analysis Toolkit (GATK v4.0.3). The PathSeq analysis was performed using prebuilt human and microbial reference genomes from the PathSeq resource bundle, available through the Broad Institute's Genome Sequence Archive (GSA) FTP server. All sequencing data were analyzed using a local high-performance computing cluster. H. pylori status of a sample was determined by having at least 1 unambiguous read in either WGS or RNAseq from the tumor tissue or either WGS, WES, or RNAseq from normal adjacent tissue.

**Statistics.** All statistical analysis was performed using Graph Pad prism 5.0 software. One-way ANOVA was used for statistical comparisons of groups of unequal sizes, followed by Dunn's multiple comparisons correction. Two-way ANOVA with Tukey's multiple comparisons correction was used wherever group sizes were identical. P-values are indicated as follows: *, <0.05; **, <0.01; ***, <0.005; ****, <0.001.

**Reporting summary.** Further information on research design is available in the Nature Research Reporting Summary linked to this article.

## Data availability

The source data underlying Figs. 1–7 and Supplementary Figs. 1–7 are provided as source data files (Supplementary Data 1–13). All the other data supporting the findings of this study are available within the article and its Supplementary Information files. A Reporting Summary for this article is available as a Supplementary Information file. Access to TCGA gastric cancer data is available through the link: https://gdc.cancer.gov/about-data/publications/stad_2014.

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

## Acknowledgements

This study was funded by the Swiss National Science Foundation (BSCGIO 157841/1 to A.M. and 310030_184716 to P.J.). Additional support was provided by the Swiss Cancer League (KFS-3802-02-2016 to P.J.), the Czech Science Foundation (19-07674S to P.J.), and the Czech Ministry of Education, Youth and Sports (LTAUSA19096 to J.D.). We are grateful to Massimo Lopes for advice and discussions. We thank the UZH Center for Microscopy and Image Analysis and the Light Microscopy Core Facility of IMG (CZ.02.1.01/0.0/0.0/ 16_013/0001775, LO1419, LM2018129, CZ.2.16/3.1.00/21547) for support.

## Author contributions

M.B., Z.N., A.M., and S.L. performed experiments and analyzed the data. M.J. and S.B. planned and performed gastric organoid experiments together with M.B.; A.G., N.R.S., R.H., M.S., L.P., J.D., and T.F.M. provided critical reagents, cell lines, and bacterial mutants. J.D. supervised. Z.N., A.T., and A.W. contributed human gastritis samples and performed IHC experiments. P.F.C. performed TCGA data analysis with input from M.P.L. and A.W. A.D. and X.S. annotated TCGA data with the *H. pylori* status. P.J. and A.M. conceived the study, analyzed the data, and wrote the manuscript.

## Competing interests

The authors declare no competing interests.
