## [Peer Review File · Nature Communications]

Reviewers' comments:

Reviewer #1 (Expertise: DNA damage and R loops, Remarks to the Author):

This is a nice manuscript in which authors use the known fact that *Helicobacter* causes DSBs in gastric epithelial cells to show that such breaks occurs in S-phase in a co-transcriptional manner. By studying whether the beta-ADP-heptose/ALPk1/TIFA signaling path has any role in DNA damage induction, they first show that ablation of the path reduces damage as determined by 53BP1 foci. Since ALPk1/TIFA activates NF-kappaB activation, authors explore whether damage is transcription dependent and they show it via inhibition of transcription with triptolide. Further exploration permits the author to see that damage is only observed in PCNA+ cells and that is replication-dependent to finally provide data using RNH1 to propose that this stress is regulated by R loop accumulation. The study is certainly interesting and well executed, even though at the end it may just represent another example of the impact of RNA hybrids in replication stress and replication-dependent DNA damage. This said, few aspects should be fixed before the paper is considered for publication, including a more complete analysis of DNA-RNA hybrids, a much better list of references and a modification and tuned down of the conclusions related to cancer and obtained from the database analysis.

- Authors represent length of tracks from the fiber analysis. Why not to show directly speeds? This is important to relate with other publications and to really see what is the difference observed and how is is suppressed by RNH.

- Authors should not talk about replication stalling. Apparently, they show reduction in replication speed, but in order to talk about stalling they show analyze fork asymmetry

- Plots in Fig. 5 and 6 should be rearranged so that we can see directly the data from +DOX and -DOX close by for each condition. It is the statistical analysis between both treatments what really matters. It is not obvious whether there is a significant suppression of tracts and 53BP1 foci, not only because the way the plot looks like but because there is no statistic performed on this comparison or it is non significant, but is not said in the plot.

- Given the confirmatory or supportive role of this manuscript in R loops with replication stress and fork conflicts as a source of damage authors should refer the previous bibliography in yeast and human cells from the Aguilera, Cimprich and Merrikh labs minimally.

- The only way authors address directly the identification of hybrids is by using a mutated RNH1 in vivo. This is not sufficient, because authors need to remove the signal treating cells afterwards with active RNH1 to demonstrate that the signal is specific. This is not easy when the treatment are done in vivo. Thus, authors need to validate at least a series of experiments using S9.6 and removing the signal with RNH1 in vitro. In this sense, IF studies on R loops are always difficult to interpret and should be accompanied by DRIP-qPCR analysis at specific genes (authors have a good list of genes that are regulated by the conditions used), in which case they will be able to quantify the signal by PCR and remove it with in vitro RNH1 treatment. In this case this analysis is particularly important because RNH1 is one of the enzymes that removes the RNA primer of Okazaki fragments and authors are studying a replication-linked phenomenon, according to their conclusions.

- The use of CPT as a control is nice, but the meaning is unclear to me. CPT acts in a completely different manner since it causes TOP1cc, which is behind damage and replication impairment regardless of transcription. It is not either a good control for DNA-RNA hybrids, among other reasons because it is not the same to remove TOP1 than to use CPT. These limitations should be stated and the way the CPT data is used as a confirmation of the conclusions should be tuned down thru the text.

- To confirm that replication stress is caused by DNA-RNA hybrids dependent on transcription rather than a problem with Okazaki removal, authors should be able to suppress the replication stress observed with triptolide.

- The last session on the relationship between gastric cancer and R loops should be reduced and conclusions tuned down. The observation that replication stress and DNA damage is abundant in cancer cells and that this is in many cases accompanied by mutations in DDR factors and RNA processing factors is not new. This manuscript does not add anything new on this. My concern is that they use the information in a misleading manner leading to over-conclusions not supported at this moment. Thus, the list of genes that is selected in Discussion is just a minor proportion of the long list of genes mutated. Those discussed are not that relevant in many cases given the number of mutations encountered. In some cases, it is not even obvious why authors include some of the mutations in their rational. This is the case of TREX1 and TREX2. These are two exonucleases. What are the previous reports showing a relationship of these genes with R loop? Are the authors confusing these genes with the protein complexes TREX and TREX-2 that have nothing to do?

- In general, this manuscript shows a poor reference citations. Thus, authors indicate that SIN3A, TOP3B, PIF1 and other are involved in R loops. If so, references need to be cited. In this sense, it is interesting that some of the paradigmatic genes involved in R loop, like THOC2 that appears in the list, or even SRSF1, are not even mentioned in the text to support the argument. But they make a big case of TREX2 together with PIF1 indicating that they control R loops without adding the references.

It seems authors did not make sufficient to stress the relevance and difference in action of these genes.

- Reference PMID: 26584049 should be cited together with ref 46. Both reported the same conclusions and were published the same month.

Reviewer #2 (Expertise: T4 Secretion system, Remarks to the Author):

H. pylori (Hp) infection is a significant risk factor for gastric carcinogenesis. Upon infection, *H. pylori* delivers the metabolite ADP-beta-D-manno-heptose (β -ADP-heptose) via its Cag T4SS to gastric epithelial cells, which activates NF- κ B signaling through the sensor/adaptor ALPK1/TIFA. The authors have previously shown that *H. pylori* infection induces DNA double-strand breaks (DSBs) which is dependent on the functional T4SS encoded by the Cag pathogenicity island (Cag-PAI) but not dependent on the T4SS-translocated effector CagA. Hp-induced DNA damage is dependent on NF- κ B activation, and is largely restricted to transcribed regions. In this paper the authors investigated a possible link between NF- κ B activation by ADP-heptose/ALPK1/TIFA and NF- κ B induced DNA damage to detail the molecular mechanism of Hp-induced DNA damage. They report Hp-induced DNA damage occurs by co-transcriptional R-loops formation in S-phase cells upon Hp infection. This is an interesting and well-written paper, and the findings significantly advance our understanding of the causal links between *H. pylori* infection, Hp-induced DNA damage, and gastric cancers. There are just a few minor comments to address:

Minor:

1. Addition of TNF α , a canonical NF κ B activator, did not result in DNA damage even at concentrations that trigger IL-8 secretion at levels comparable to Hp infection (Fig. 2A,B,C). This result should be interpreted and discussed. Is it possible, for example, that upon Hp infection, in addition to NF κ B activation there are additional factor(s) encoded by Hp or the host necessary that act through a different pathway(s) to induce DNA damage?
2. Describe the abbreviation 'H&E' in Fig. 2D legend.
3. DSBs were observed after infection of both cag-PAI-positive and -negative HP strains, and both are also associated with gastric carcinogenesis but the effect is more robust with the cag-PAI-positive strains (Hanada et al 2014 82:4182-89, Infection and Immunity). This information should be presented in the Introduction, and discussed further in relation to the present findings; basically, a functional cag T4SS is not critical for Hp-induced DSBs, correct?

4. Pg. 8, L. 10. In light of the above, it would be useful to see a direct comparison of DSBs by PFGE upon infection by Hp cag-PAI -negative and -positive strains in Fig. 3 or supp Fig 3, rather than simply referencing the finding (#18).

5. Fig. 3. Perhaps I missed this, but DSBs are not shown for CTRL cells supplemented with α ADP-heptose and β ADP-heptose, are they? If not, this should be shown.

6. In lines 5 and 6 of page 8, replace supp. Fig. 3a with 3ab.

In line 8 of page 8, replace supp. Fig 3b with 3c.

In line 15 of page 8, replace supp. Fig 3c with 3b.

7. Fig. 4C. It appears that there is a further decreased replication stress for Hp WT infection of the TIFA mutant compared with the ALPK1 mutant. Also, in the case of β ADP-heptose addition for these two mutants. Are these statistically significant differences, if so, what are the authors' thoughts? Could there be an additional ALPK1-independent but TIFA-dependent mechanism for promoting DNA replication stress upon Hp infection?

8. Figs. 4E and F. Addition of TNF α suppressed replication stress significantly, but this is not described in the text. Describe and interpret.

9. Pg. 11, L. 1-3. The "data not shown" should be shown in a supplemental Fig.

10. P. 13, L. 11-12. It is mentioned that the authors identified 17 genes with functions of R-loop prevention and processing through a literature search. These references should be cited.

11. P. 16, L. 17-21. Hp infection is linked to genomic instability via reactive oxygen and nitrogen species reviewed by Kidane D (International Journal of Molecular Sciences 2018, 19:2891). Although the authors mention ways by which R-loops might be formed upon Hp infection, they do not experimentally address or entertain the possibility that damaged nucleotides by reactive oxygen and nitrogen species contribute to R-loops. It is conceivable that replication fork blockage at lesion sites in the sense strand inflicted by reactive oxygen/nitrogen species can cause the accumulation of transcripts from the antisense strand and lead to formation of R-loops.

Reviewer #3 (Expertise: Gastric cancer, H.Pylori, Remarks to the Author):

Bauer et al. show that H. pylori generates DSBs (53BP1 foci) in S-phase cells by activating the NF- κ B transcription factor in a manner that is dependent on the T4SS/ALPK/TIFA/NF- κ B signaling axis, which was recently identified by several groups including the authors' group (Zimmermann et al. Cell Reports 20, 2384-2395, 2017; Gall et al. mBio 8, e011687-17, 2017; Stein et al. PLoS Pathog 13, e1006514, 2017). The authors also show that H. pylori infection-mediated DSBs are due to

accelerated R-loop formation in NF- κ B-regulated genes, which induces replication fork stress or by cutting the R-loop with the nucleotide excision repair enzymes such as XPG or XPF as reported previously by the authors' group (Hartung et al. Cell Reports 13, 70-79, 2015).

Although the experiments were well-designed and the results were nicely demonstrate, my overall impression after reading the manuscript is that it is made by putting together two works previously published by the authors' group [(Hartung et al. Cell Reports 13, 70-79, 2015) and (Zimmermann et al. Cell Reports 20, 2384-2395, 2017)] by arguing that increase in transcriptional activity promotes R-loop formation and subsequent replication stress. Indeed, it has already been shown that the nucleotide excision repair endonucleases XPG and XPF promote R-loop formation and thereby induce DSBs. Since most of the molecular mechanisms involved in DSB induction following *H. pylori* infection presented in this work were reported or discussed in their previous works (Hartung et al. 2015 and by Zimmermann et al. 2017), I cannot find any novel findings/discoveries or new conceptual breakthroughs in the present study.

The title of the paper "The ALPK1/TIFA/NF- κ B axis links a bacterial carcinogen to R-loop-induced replication stress" argues that their finding is associated with the cancer development. Although genomic instability has been shown to play an important role in carcinogenesis, the present study does not describe how it could promote gastric carcinogenesis by *H. pylori* (including the merit of cancer development for *H. pylori*)? Especially, host gastric epithelial cells receiving massive DNA damages (DSBs) should undergo apoptosis and thus excluded from the body unless repaired by an extremely efficient way. If the *H. pylori*-damaged cells could survive despite severe DSBs, why and how? I think that this question needs to be clarified in order to argue the relationship between *H. pylori*-induced DNA damage and carcinogenesis.

It is also weird why activation of NF- κ B other than the non-ALPK1/TIFA axis is incapable of inducing DSBs in the host cells. The finding suggests that activation of NF- κ B is required but not sufficient for induction of DSBs and suggests the involvement of additional *H. pylori*-host cell interaction. This possibility could be tested by treatment of AGS cells that had been infected with *H. pylori* strains lacking RfaE or T4SS with TNF α .

The followings are other specific comments:

1. In Figs. 1, 3, and 4, the authors used ALPK1- or TIFA-knockout AGS cells established by a CRISPR/Cas9 technology. The each knockout cell line was generated by using a single guide RNA sequence. To rule out off-target effects, multiple targeting sequences are required for the same target. The authors should reproduce the data using multiple knockout cell lines.

2. In Fig. 2d, they should show that γ H2AX and 53BP1 staining relative to H. pylori staining in serial sections.

3. In Fig. 7c, they analyzed the mutations of the R-loop genes in the gastric cancer genomic data reported by the Cancer Genome Atlas (TCGA). However, they did not show that correlation between the mutations and H. pylori infection.

Point by Point response to reviewers

We would like to thank our reviewers for constructive comments, which we have now done our best to address experimentally throughout. In addition to the experiments requested by individual reviewers, we have added data on three major topics to the manuscript:

Most importantly, we have performed a much more extensive bioinformatic analysis of the TCGA data. In particular, we have collaborated with additional bioinformaticians at Duke Univ (now listed as new co-authors) who specialize in mining non-human sequences in TCGA data for the presence of microbial DNA and transcripts. This novel and very exciting approach has allowed us to annotate the TCGA samples with their *H. pylori* status and to investigate in detail whether the mutational patterns differ in each of the four subtypes of gastric cancer as a function of *H. pylori* presence; indeed, we found that the subtypes “microsatellite instable” (MSI^{hi}) and “chromosomally instable”(CIN) exhibit a higher mutational burden if *H. pylori* is present, pointing to genomic aberrations that are related to *H. pylori*. Indeed, the genes affected by mutations, certainly in the MSI^{hi} subtype, seem to differ as a consequence of the presence or absence of *H. pylori*.

A second addition to the Results section is based on the cytokinesis block micronucleus assay, which allows for the sensitive and quantitative assessment of genotoxicity. Interestingly, we found that *H. pylori* robustly induced micronuclei formation in an RfaE-dependent manner, which is perfectly consistent with the idea that replication stress occurring in S phase results in consequences during mitosis, jeopardizing chromosome segregation and, in turn, genome stability. Along the same line, we have now added data showing that, despite the massive DNA damage incurred, *H. pylori* infected cells survive and fail to launch a DNA damage response as judged by lack of ATM and KAP-1 phosphorylation. This latter finding explains the conundrum of why *H. pylori*-infected cells survive despite massive damage to their genome.

We also attempted to provide direct evidence for a mutational signature of *H. pylori* in AGS cells. To this end, we repeatedly (within three cycles) exposed AGS cells to *H. pylori* and then killed the bacteria by antibiotics to allow the cells to recover and repair their DNA damage. We were hoping that these three cycles of two days each would already lead to a detectable accumulation of *H. pylori*-induced mutations. However, whole genome sequencing of 10 clones (derived from single cells) of *H. pylori*-exposed AGS cells did not reveal any consistent evidence of point mutations or indels that would have been directly attributable to the bacterial exposure. We interpret these data to mean that much more chronic (i.e. over years and decades, not days) exposure to *H. pylori* is probably needed to manifest in the form of mutations, as is evident from the TCGA data.

All other points raised by our reviewers were addressed experimentally wherever possible and are described in detail below (answers in red).

Reviewers' comments:

Reviewer #1 (Expertise: DNA damage and R loops, Remarks to the Author):

This is a nice manuscript in which authors use the known fact that Helicobacter causes DSBs in gastric epithelial cells to show that such breaks occurs in S-phase in a co-transcriptional manner. By studying whether the beta-ADP-heptose/ALPk1/TIFA signaling path has any role in DNA damage induction, they first show that ablation of the path reduces damage as determined by 53BP1 foci. Since ALPk1/TIFA activates NF-kappaB activation, authors explore whether damage is transcription

dependent and they show it via inhibition of transcription with triptolide. Further exploration permits the author to see that damage is only observed in PCNA+ cells and that is replication-dependent to finally provide data using RNH1 to propose that this stress is regulated by R loop accumulation. The study is certainly interesting and well executed, even though at the end it may just represent another example of the impact of RNA hybrids in replication stress and replication-dependent DNA damage. This said, few aspects should be fixed before the paper is considered for publication, including a more complete analysis of DNA-RNA hybrids, a much better list of references and a modification and tuned down of the conclusions related to cancer and obtained from the database analysis.

- Authors represent length of tracks from the fiber analysis. Why not to show directly speeds? This is important to relate with other publications and to really see what is the difference observed and how is suppressed by RNH.

Response: This is a valid point, although our feeling is that some researchers prefer to show tract length, whereas others prefer to show speed. To do justice to everyone, we are now showing both length and speed for two key experimental setups (three experiments each). One is fork slowing induced by *H. pylori* infection or β -heptose treatment in WT, but not TIFA- or ALPK1-deficient AGS cells; the other is fork slowing in U2OS cells that were infected with wild type bacteria, and treated or not with doxycycline to induce RNase H1 (please see suppl. Fig. 4a and suppl. Fig. 5c).

The graphs showing lengths and speeds, not surprisingly, look exactly the same, but the y-axis and unit differs of course. The text and figures have been changed accordingly as follows:

Results, p.9:

We found that *H. pylori* infection led to slowing of replication fork progression in wild type AGS cells as judged by measuring the lengths of CldU tracts (Fig. 4b,c) and also by plotting replication fork speeds (suppl. Fig. 4a) that were calculated based on the assumption that 1 μ m of fiber corresponds to 2.59 kb.³⁴

and

Results, p.10:

Exposure of U2OS cells to wild type *H. pylori* for six hours induced the formation of 53BP1 foci that increased with the MOI and were specific to S-phase, and were comparable in extent to those observed in AGS cells (Fig. 5a,b, left panels -DOX, suppl. Fig. 5a,b). This DNA damage phenotype was accompanied by replication fork slowing as determined by DNA fiber assay (Fig. 5c,d, suppl. Fig. 5c).

Importantly, the induction of RNase H1 expression by doxycycline abrogated both 53BP1 foci formation (Fig. 5a,b, right panels, +DOX) and replication fork slowing (Fig. 5c,d, lower and right panels, +DOX, suppl. Fig. 5c) upon *H. pylori* infection, and also upon β -ADP-heptose treatment (Fig. 5e-h).

- Authors should not talk about replication stalling. Apparently, they show reduction in replication speed, but in order to talk about stalling they show analyze fork asymmetry

Response: We have now analyzed fork asymmetry in detail for the two most important setups, i.e. the infection of wild type AGS cells with wild type *H. pylori* and the infection of U2OS cells with *H. pylori*, with and without DOX-induced expression of RNase H1. These analyses indeed revealed that replication fork stalling happens in the presence of *H. pylori*, and this can be reversed by RNase H1.

Both pieces of data are now included in the supplements to Figures 4 and 5. Nevertheless, we now make sure to use the term “fork slowing” instead of “fork stalling” throughout (except for the chapters where we talk about fork asymmetry), to avoid over-interpretation of the data. The entire text has been corrected in this manner. The text on fork asymmetry has been added as follows:

Results, p.9: Ongoing replication events were sequentially labeled with two thymidine analogs—chlorodeoxyuridine (CldU) and iododeoxyuridine (IdU)—and individual two-color labeled DNA tracts were visualized on stretched DNA fibers by immunofluorescence microscopy (Fig. 4a,b). We found that *H. pylori* infection led to slowing of replication fork progression in wild type AGS cells as judged by measuring the lengths of CldU tracts (Fig. 4b,c) and also by plotting replication fork speeds (suppl. Fig. 4a) that were calculated based on the assumption that 1 μm of fiber corresponds to 2.59 kb.³⁴ As replication fork slowing and shorter tracts can, in this assay, result from two scenarios, i.e. either a slower overall DNA polymerization rate or increased frequency of fork stalling,³⁵ we analyzed the fates of two (sister) replication forks emanating in opposite directions from the same origin. To this end, we calculated the ratio of the lengths of CldU tracts of sister replication forks (length of the shorter tract divided by the length of longer tract); in uninfected cells, the ratio was ~ 1 , indicating that sister forks traveled at similar speed. In *H. pylori*-infected AGS cells, the ratio dropped to ~ 0.6 , indicating fork asymmetry and selective slowing/stalling of one fork only (suppl. Fig. 4b).

and

Results, p.12: The analysis of sister forks emanating from the same origin showed that not only the slowing of fork progression, but also the fork asymmetry associated with *H. pylori* infection, could be reversed by induction of RNase H1 (suppl. Fig. 5g,h). The combined results implicate R-loops, processed by RNase H1, in DNA damage and replication stress induced by *H. pylori*.

- Plots in Fig. 5 and 6 should be rearranged so that we can see directly the data from +DOX and -DOX close by for each condition. It is the statistical analysis between both treatments what really matters. It is not obvious whether there is a significant suppression of tracts and 53BP1 foci, not only because the way the plot looks like but because there is no statistic performed on this comparison or it is not significant, but is not said in the plot.

Response: We tried to arrange the plots in Figures 5 and 6 so that the – and + DOX conditions are presented next to each other. However, these plots were now EVEN MORE difficult to read. We therefore resorted to a color code, where the – DOX and + DOX samples of the same condition share the same color. We think this system works well, and hope our reviewer agrees. Below are two examples.

- Given the confirmatory or supportive role of this manuscript in R loops with replication stress and fork conflicts as a source of damage authors should refer the previous bibliography in yeast and human cells from the Aguilera, Cimprich and Merrikh labs minimally.

Response: Several new references from the three mentioned labs are now included in various sections of the text, as correctly requested by this reviewer.

Results section (key review articles):

R-loop formation is required for *H. pylori*-induced replication stress and DNA damage

Active replication and transcription that co-occur in the same regions of the genome can result in replication stress and DNA damage if both machineries collide. Replication fork stalling at sites of these conflicts is caused by the formation of co-transcriptional R-loops that represent a potent block to replication fork progression.³⁶⁻³⁸

36. Crossley, M.P., Bocek, M. & Cimprich, K.A. R-Loops as Cellular Regulators and Genomic Threats. *Molecular cell* **73**, 398-411 (2019).
37. Aguilera, A. & Garcia-Muse, T. R loops: from transcription byproducts to threats to genome stability. *Molecular cell* **46**, 115-124 (2012).
38. Garcia-Muse, T. & Aguilera, A. R Loops: From Physiological to Pathological Roles. *Cell* **179**, 604-618 (2019).

Discussion (the original literature): p.18ff

As shown by multiple laboratories in various models systems -ranging from bacteria, to yeast, to mammalian cells- R-loops may either form as a consequence of head-on collisions between a replication fork and transcription bubble as proposed previously, or alternatively, may form in highly transcribed regions of the genome, where they cause pausing of RNA-Pol II, which in turn blocks the progression of oncoming replication forks.^{36, 49-52} Irrespective of the sequence of events, we find that R-loop accumulation upon *H. pylori* exposure promotes S-phase-specific DNA damage, as overexpression of RNase H1 reduces DNA damage as judged by pulsed-field gel electrophoresis as well as 53BP1 immunostaining, and abrogates replication stress. Both properties of RNase H1 require its enzymatic activity and are not observed with a catalytically dead version of the enzyme. R-loops have been linked not only to DNA damage, genomic instability and chromosomal rearrangements, but also to oncogenesis^{50, 53-56}. Increased transcriptional activity resulting from oncogene activation is known to promote R-loop accumulation and replication stress.

36. Crossley, M.P., Bocek, M. & Cimprich, K.A. R-Loops as Cellular Regulators and Genomic Threats. *Molecular cell* **73**, 398-411 (2019).
- ...
49. Allison, D.F. & Wang, G.G. R-loops: formation, function, and relevance to cell stress. *Cell Stress* **3**, 38-46 (2019).
50. Garcia-Rubio, M.L. *et al.* The Fanconi Anemia Pathway Protects Genome Integrity from R-loops. *PLoS genetics* **11**, e1005674 (2015).
51. Lang, K.S. *et al.* Replication-Transcription Conflicts Generate R-Loops that Orchestrate Bacterial Stress Survival and Pathogenesis. *Cell* **170**, 787-799 e718 (2017).
52. Herrera-Moyano, E., Mergui, X., Garcia-Rubio, M.L., Barroso, S. & Aguilera, A. The yeast and human FACT chromatin-reorganizing complexes solve R-loop-mediated transcription-replication conflicts. *Genes & development* **28**, 735-748 (2014).
53. Schwab, R.A. *et al.* The Fanconi Anemia Pathway Maintains Genome Stability by Coordinating Replication and Transcription. *Molecular cell* **60**, 351-361 (2015).

54. Helmrich, A., Ballarino, M. & Tora, L. Collisions between replication and transcription complexes cause common fragile site instability at the longest human genes. *Molecular cell* **44**, 966-977 (2011).
55. Kotsantis, P. *et al.* Increased global transcription activity as a mechanism of replication stress in cancer. *Nature communications* **7**, 13087 (2016).
56. Bhatia, V. *et al.* BRCA2 prevents R-loop accumulation and associates with TREX-2 mRNA export factor PCID2. *Nature* **511**, 362-365 (2014).

- The only way authors address directly the identification of hybrids is by using a mutated RNH1 in vivo. This is not sufficient, because authors need to remove the signal treating cells afterwards with active RNH1 to demonstrate that the signal is specific. This is not easy when the treatment are done in vivo. Thus, authors need to validate at least a series of experiments using S9.6 and removing the signal with RNH1 in vitro. In this sense, IF studies on R loops are always difficult to interpret and should be accompanied by DRIP-qPCR analysis at specific genes (authors have a good list of genes that are regulated by the conditions used), in which case they will be able to quantify the signal by PCR and remove it with in vitro RNH1 treatment. In this case this analysis is particularly important because RNH1 is one of the enzymes that removes the RNA primer of Okazaki fragments and authors are studying a replication-linked phenomenon, according to their conclusions.

Response: We include a set of experiments that were stained with the S9.6 antibody, which we find a useful tool to stain R-loops in cultured cells.

Results, p.12:

Interestingly, the induction of RNase H1^{D210N}/GFP foci and of DNA damage correlated strongly with the level of incorporation of 5-fluorouracil (suppl. Fig. 5c), which was also strongly dependent on *rfaE*, the Cag-PAI and NF- κ B, and which we used as an indicator of RNA synthesis. **Staining of infected cells with an antibody specifically recognizing RNA/DNA hybrids (clone S9.6) confirmed that infection with wild type *H. pylori*, but not its *rfaE* and Cag-PAI mutants, induces R-loops not only in U2OS cells as shown with the reporter cell line, but also in AGS cells (suppl. Fig. 6e,f). Taken together,...**

We have more recently also attempted to use extracted DNA in a dot plot strategy, in combination with the S9.6 antibody, to examine whether induction of RNaseH1 indeed reduced the S9.6 signal. This was indeed the case as shown representatively for the β -heptose induced R-loop formation. However, the dot blot method needs optimization and we don't feel confident enough using this strategy to actually show it as a panel in the manuscript.

For this dot blot, extracted DNA from α -heptose (negative control) and β -heptose-treated USOS cells, treated or not to induce RNaseH1, was stained with either S9.6 antibody, or methylene blue as loading control. It is quite clear that RNaseH1 strongly reduces the S9.6 signal, and that only the β - but not the α -heptose induces it. The dot blot result is thus consistent with the RNaseH1^{GFP} reporter cell line results.

We also performed DRIP-qPCR analysis for the *IL8* gene, but despite numerous attempts, were not able to get the system to work for this revision. We feel that a genome-wide chromatin-IP-based survey of RNaseH1-bound genome regions would be the appropriate way to address where R-loops form in the genome of *H. pylori*-exposed cells. DRIP-qPCR for at least a few loci would have been really nice to have, but sadly, we can't provide this data due to technical difficulties.

- The use of CPT as a control is nice, but the meaning is unclear to me. CPT acts in a completely different manner since it causes TOP1cc, which is behind damage and replication impairment regardless of transcription. It is not either a good control for DNA-RNA hybrids, among other reasons because it is not the same to remove TOP1 than to use CPT. These limitations should be stated and the way the CPT data is used as a confirmation of the conclusions should be tuned down thru the text.

Response: As camptothecin was questioned by this reviewer as the proper positive control, we are now also showing a second positive control, the G-quadruplex ligand pyridostatin in the DNA fiber assay. G4 ligands are known to induce DNA damage and genome instability, and indeed, we confirmed this also in AGS gastric epithelial cells. The quantification of pyridostatin-induced fork slowing is shown in the new Fig. 4d, and mentioned in the text as follows:

Results, p.9: The effect size of *H. pylori* infection on replication fork progression was comparable to the effects of the topoisomerase inhibitor camptothecin,³⁵ and to the effects of the G quadruplex DNA (G4) ligand pyridostatin, a well-known inducer of both DNA damage and genome instability³⁶, which served as our positive controls (Fig. 4d).

We are also now more careful to not overstate the usefulness of camptothecin, although we have been advised by our colleagues at IMCR that this is indeed a widely used control for both R-loop formation and DNA damage.

Although we tried to directly measure DNA damage as 53BP1 foci upon pyridostatin treatment, these experiments failed as the immunofluorescence staining was somehow (in multiple attempts) compromised (i.e. much too bright) in all samples to which we had added the pyridostatin. Despite trying hard, we could not overcome this problem.

It should also be noted that, in our recently published work we showed that camptothecin- and pyridostatin-induced replication fork slowing is caused by co-transcriptional R-loops in U2OS cells (Chappidi et al. Molecular Cell 2020).

- To confirm that replication stress is caused by DNA-RNA hybrids dependent on transcription rather than a problem with Okazaki removal, authors should be able to suppress the replication stress observed with triptolide.

Response: This is indeed the case! The replication stress as read out by fiber shortening is completely reversible by triptolide, and also by the inhibition of NF- κ B-driven transcription with BAY11. The data was in fact already in the first version of the manuscript, but not sufficiently mentioned in the text. Also, the statistical evaluation was missing for the comparison of fiber length upon infection with and without triptolide in the U2OS setting. We have now added the respective p-values to the graph in question in suppl. Figure 5e, and added the following statements to the text:

Results, p.10:

Exposure of infected cells to the NF- κ B inhibitor BAY 11-7082 or to the transcription inhibitor triptolide rescued *H. pylori*-induced fork slowing (Fig. 4e,f), suggesting that active transcription driven by this

transcription factor as a consequence of β -ADP-heptose delivery and ALPK1/TIFA signaling is a prerequisite of replication stress in *H. pylori*-infected cells.

and

Results. p.10:

This DNA damage phenotype was accompanied by replication fork slowing as determined by DNA fiber assay (Fig. 5c,d, **suppl. Fig. 5c**). As observed for AGS cells, replication fork slowing in *H. pylori*-infected U2OS cells was dependent on a functional Cag-PAI and RfaE, **and could be completely reversed by blocking transcription with the inhibitor triptolide or the NF- κ B inhibitor BAY 11-7082 (suppl. Fig. 5d,e)**. Both DNA damage in S-phase cells and replication stress could be induced also in U2OS cells by the addition of β - but not α -ADP-heptose (Fig. 5e-h, upper and left panels, -DOX, **suppl. Fig. 5f**). Importantly, the induction of RNase H1 expression by doxycycline abrogated both 53BP1 foci formation (Fig. 5a,b, right panels, +DOX) and replication fork **slowing** (Fig. 5c,d, lower and right panels, +DOX, **suppl. Fig. 5c**) upon *H. pylori* infection, and also upon β -ADP-heptose treatment (Fig. 5e-h). **The analysis of sister forks emanating from the same origin showed that not only the slowing of fork progression, but also the fork asymmetry associated with *H. pylori* infection, could be reversed by induction of RNase H1 (suppl. Fig. 5g,h)**. The combined results implicate R-loops, processed by RNase H1, in DNA damage and replication stress induced by *H. pylori*.

- The last session on the relationship between gastric cancer and R loops should be reduced and conclusions tuned down. The observation that replication stress and DNA damage is abundant in cancer cells and that this is in many cases accompanied by mutations in DDR factors and RNA processing factors is not new. This manuscript does not add anything new on this. My concern is that they use the information in a misleading manner leading to over-conclusions not supported at this moment. Thus, the list of genes that is selected in Discussion is just a minor proportion of the long list of genes mutated. Those discussed are not that relevant in many cases given the number of mutations encountered. In some cases, it is not even obvious why authors include some of the mutations in their rational. This is the case of TREX1 and TREX2. These are two exonucleases. What are the previous reports showing a relationship of these genes with R loop? Are the authors confusing these genes with the protein complexes TREX and TREX-2 that have nothing to do?

Response: We agree with this reviewer that R-loop processing genes are just a subset of a long list of genes that are mutated in gastric cancer. We have moved this figure to the supplement, and have toned down all text sections related to its content. The TREX exonucleases have been removed; instead, we have looked at the components of the TREX-2 complex, as suggested, which showed a very high extent of CNVs and turned out to be among the most interesting genes in the analysis.

The text has been modified as follows:

Results, p.14: We also looked specifically for mutations and copy number variations (CNVs) in genes involved in R-loop prevention and processing, with the idea that mutations in such genes might predispose to gastric carcinogenesis. We plotted the mutations and CNVs in **18 genes** with functions in R-loop prevention or processing relative to seven known gastric cancer driver genes, i.e. *CDH1*, *APC*, *TP53*, *ARID1A*, *PIK3CA*, *KRAS* and *ERBB2*, and relative to the mutational burden per megabase. **The 18 genes with R-loop metabolism-related functions were selected based on literature searches: we focused on the RNA/DNA helicases *AQR*, *SETX* and *DHX9* involved in R-loop unwinding,^{19, 43, 44} the DNA helicase *PIF1* known to prevent the accumulation of R-loops at tRNA genes,⁴⁵ the DNA topoisomerase *TOP3B*, which also prevents accumulation of R-loops,⁴⁶ the endonuclease *RNASEH2* which, like RNase H1 cleaves the RNA strand in RNA/DNA hybrids,⁴⁷ the splicing factor *SRSF1*, known to prevent R-loop**

formation,⁴⁸ *THOC1* and *THOC2*, components of the THO/TREX complex involved in mRNA export and R-loop prevention,⁴⁹ the THO/TREX complex associated histone deacetylase *SIN3A*,⁵⁰ and the six components of the TREX-2 complex that, with *BRCA2*, is also associated with mRNA export and R-loop prevention.⁵¹ Several of these genes were recurrently (in up to 36% of gastric cancer cases) affected by CNVs and/or missense or frameshift mutations, with the most recurrently affected genes belonging to the TREX-2 complex (*ENY2*, *MCM3AP*, *SEM1*, *CETN3*) or having RNA/DNA helicase or endonuclease activity (*SETX*, *AQR*, *DHX9*, *RNASEH2*) (suppl. Fig. 7c). Mutations in R-loop metabolism genes were largely predicted to be damaging mutations, and -not unexpectedly- were mostly detected in the microsatellite-unstable subtype of gastric cancer that exhibited the highest overall mutational burden; CNVs affecting R-loop genes were mostly detected in the chromosomally unstable subtype (suppl. Fig. 7c). In conclusion, the mutational landscape of gastric cancer provides circumstantial evidence for the hypothesis that *H. pylori* infection favors chromosomal instability, and that the inactivation of genes encoding helicases, RNases and other factors involved in R-loop prevention or processing may be an early event in gastric carcinogenesis, predisposing *H. pylori*-infected cells to aberrant R-loop accumulation and its potentially deleterious consequences.

Response: In addition, we are now including data in the main Figure 7 that are more relevant to the topic. Through help from two bioinformaticians specializing in annotating non-human sequences found in TCGA data to specific microbes, Anders Dohlman and Xiling Shen from Duke University (who now figure as co-authors), we were able to annotate the *H. pylori* status to all of our gastric cancer samples. This annotation, which is based on the presence of *H. pylori* transcripts or genomic sequencing in either the tumor or its adjacent tissue (where accessible), has allowed us to compare the tumor mutational burden as a function of *H. pylori* presence. Interestingly, this analysis showed that gastric cancers from *H. pylori*-infected patients (of the most common MSI^{hi} and CIN subtypes) tend to have a higher mutational burden. Over 200 genes are differentially mutated as a function of *H. pylori* status. These genes, along with a quantification of tumor mutational burden (TMB) per subtype, is now shown in Figure 7 along with the following explanatory text.

Results, p.13:

***H. pylori* is associated with a higher mutational burden in two subtypes of gastric cancer**

We next asked whether gastric cancer genomic data would possibly provide circumstantial evidence for the presence of *H. pylori* as a driving force of malignant transformation. To this end, we used the non-human transcript and genomic information that is present in publicly accessible multi-omics (RNA-sequencing, whole genome sequencing and whole exome sequencing) datasets of gastric cancer, which had been generated as part of the Cancer Genome Atlas (TCGA) project, to annotate gastric cancer samples with their *H. pylori* status. We had access to multi-omics datasets for 291 tumors from treatment-naïve gastric cancer patients that had previously been subjected to array-based somatic copy number analysis, whole-exome sequencing and array-based DNA methylation profiling, and that had resulted in the description of four major genetically defined subtypes of gastric cancer.⁴² These subtypes were EBV-positive gastric cancer (characterized by presence of EBV genes and transcripts, a very low mutational burden and *PIK3CA* mutations), microsatellite-unstable gastric cancer (high mutational burden, MSI high), chromosomally unstable gastric cancer (with large numbers of copy number variations and a high mutational burden, and ubiquitous TP53 mutations) and genomically stable gastric cancer (low mutational burden, diffuse type by histology, early onset, *RHOA* and *ARHGAP6/26* somatic genomic alterations).⁴² We were able to recapitulate the stratification of the 291 patients into these subtypes (Fig. 7e). We found evidence of *H. pylori* presence in all four subtypes, with 1/3 to 1/2 of patients of each subtype exhibiting evidence of *H. pylori* transcripts or genomic DNA in either their tumor and/or adjacent tissue (Fig. 7e). When comparing the tumor

mutational burden of the four subtypes, we found that tumors from patients with evidence of *H. pylori* infection (in either tumor and/or adjacent tissue) of the most common MSI and CIN subtypes had a higher mutational burden than those without *H. pylori* infection: within the CIN subtype, which with 136 samples was the most common of the four gastric cancer subtypes in the TCGA dataset, 57 *H. pylori*-infected patients had on average 2 +/- 1.5 mutations per megabase, whereas this number was 1.54 +/- 0.98 in the 79 *H. pylori*-negative patients (p=0.052 as determined by student's T-test). Within the MSI subtype the figures were 20 +/- 16 for 23 *H. pylori*-positive vs. 16 +/- 7 for 33 *H. pylori*-negative patients, but the difference was not statistically significant (p=0.2). A total of 232 genes were differentially affected by mutations as a function of *H. pylori* infection across all 291 samples (suppl. table 1, Fig. 7e).

- In general, this manuscript shows a poor reference citations. Thus, authors indicate that SIN3A, TOP3B, PIF1 and other are involved in R loops. If so, references need to be cited. In this sense, it is interesting that some of the paradigmatic genes involved in R loop, like THOC2 that appears in the list, or even SRSF1, are not even mentioned in the text to support the argument. But they make a big case of TREX2 together with PIF1 indicating that they control R loops without adding the references. It seems authors did not make sufficient to stress the relevance and difference in action of these genes.

Response: Please see the previous point. We have now included references to all 18 R-loop metabolism genes for which the analysis was done.

- Reference PMID: 26584049 should be cited together with ref 46. Both reported the same conclusions and were published the same month.

Response: This citation has now been included.

Discussion, p. 19:

R-loops have been linked not only to DNA damage, genomic instability and chromosomal rearrangements, but also to oncogenesis^{50, 53-56}. Increased transcriptional activity resulting from oncogene activation is known to promote R-loop accumulation and replication stress.

Please see above the list of all citations now included in this and other sections of the results and discussion.

Reviewer #2 (Expertise: T4 Secretion system, Remarks to the Author):

H. pylori (Hp) infection is a significant risk factor for gastric carcinogenesis. Upon infection, *H. pylori* delivers the metabolite ADP-beta-D-manno-heptose (β -ADP-heptose) via its Cag T4SS to gastric epithelial cells, which activates NF- κ B signaling through the sensor/adaptor ALPK1/TIFA. The authors have previously shown that *H. pylori* infection induces DNA double-strand breaks (DSBs) which is dependent on the functional T4SS encoded by the Cag pathogenicity island (Cag-PAI) but not dependent on the T4SS-translocated effector CagA. Hp-induced DNA damage is dependent on NF- κ B activation, and is largely restricted to transcribed regions. In this paper the authors investigated a possible link between NF- κ B activation by ADP-heptose/ALPK1/TIFA and NF- κ B induced DNA damage to detail the molecular mechanism of Hp-induced DNA damage. They report Hp-induced DNA damage occurs by co-transcriptional R-loops formation in S-phase cells upon Hp infection. This is an interesting and well-written paper, and the findings significantly advance our understanding of the

causal links between *H. pylori* infection, Hp-induced DNA damage, and gastric cancers. There are just a few minor comments to address:

Minor:

1. Addition of TNF α , a canonical NF κ B activator, did not result in DNA damage even at concentrations that trigger IL-8 secretion at levels comparable to Hp infection (Fig. 2A,B,C). This result should be interpreted and discussed. Is it possible, for example, that upon Hp infection, in addition to NF κ B activation there are additional factor(s) encoded by Hp or the host necessary that act through a different pathway(s) to induce DNA damage?

Response: This is an important point. Our data clearly indicate that NF- κ B activation is required, but not sufficient for DNA damage to occur in cells that actively transcribe AND replicate their DNA at the same time. We have looked at the combination of TNF α and the RfaE mutant, expecting that the two signals might synergize to induce DNA damage, but this was clearly not the case. The data is now included and discussed in the following sections:

Results, p. 8:

As in the context of live *H. pylori* infection, DNA damage upon β -ADP-heptose exposure was limited to PCNA⁺ cells (Fig. 3f, suppl. Fig. 3b,e). **The defect of the *rfaE* mutant could not be rescued by addition of TNF- α (suppl. Fig. 3f).** The combined results indicate that *rfaE* activity is required, and its product β -ADP-heptose is sufficient, to induce the ALPK1/TIFA-dependent DNA damage observed upon live *H. pylori* infection of cultured gastric epithelial cells.

Results, p.10:

The slowing of replication forks upon *H. pylori* infection was dependent on RfaE and the Cag-PAI, as both null mutants failed to cause CldU tract shortening (Fig. 4e,f). TNF- α exposure did not cause CldU tract shortening (Fig. 4e,f), **and also failed to rescue the phenotype of the RfaE mutant (suppl. Fig. 4d).**

Discussion, p.17:

Our data additionally implicate NF- κ B activation in genomic instability through co-transcriptional R-loop formation that is associated with replication fork stalling/replication stress and DNA damage. Not every pathway to NF- κ B activation is equally prone to R-loop induction and DNA damage; we have extensively tested TNF- α , which activates IKK and NF- κ B via the TRADD, TRAF2, NIK signaling axis⁵⁸ and bypasses ALPK1 and TIFA, and never found evidence of R-loops or DNA damage in this setting. **TNF- α also fails to synergize with the non-genotoxic *RfaE* mutant to induce DNA damage; these results indicate that NF- κ B signaling is required, but not sufficient, to induce DNA damage in actively transcribing/replicating cells.**

2. Describe the abbreviation 'H&E' in Fig. 2D legend.

Response: This has been done.

3. DSBs were observed after infection of both cag-PAI-positive and -negative HP strains, and both are also associated with gastric carcinogenesis but the effect is more robust with the cag-PAI-positive strains (Hanada et al 2014 82:4182-89, Infection and Immunity). This information should be

presented in the Introduction, and discussed further in relation to the present findings; basically, a functional *cag* T4SS is not critical for Hp-induced DSBs, correct?

Response: We think it is fair to say based on the existing literature that DNA DSBs are largely dependent on the Cag-PAI, which by no means contradicts the mechanism newly proposed here. The Cag-PAI-encoded T4SS on the one hand delivers CagA, which is not required for DNA damage, and on the other hand promotes the translocation of β -ADP-heptose into the cytosol of infected cells. Once present there, the β -ADP-heptose binds to ALPK1, activating it and triggering TIFAosome formation and NF- κ B activation (and the associated DNA damage). The Hanada paper, as well as two other papers looking at the Cag-PAI contribution to DNA DSBs are all cited. All reach similar conclusions. While the contribution of the Cag-PAI to DNA DSBs is not absolute, neither is its contribution to NF- κ B activation. Strains differ in their ability to Cag-PAI-independently cause a background level of NF- κ B activation, which probably explains the interstrain variation in causing DNA damage.

Introduction, p.3: We and others have demonstrated in a series of studies that *H. pylori* induces DNA double strand breaks (DNA DSBs) in gastric epithelial cells *in vitro* and *in vivo*.¹⁵⁻¹⁷ DNA DSB induction in *H. pylori*-exposed cells depends on a functional T4SS¹⁶⁻¹⁸ and preferentially occurs in transcribed regions of the genome.¹⁷ Whereas translocation of the only described protein substrate of the T4SS, CagA, does not contribute to DNA DSB induction, the depletion of NF- κ B subunits strongly reduces DNA DSB formation, suggesting that *H. pylori*-induced DNA damage is driven by active transcription of NF- κ B target genes, **which in turn is Cag-PAI-dependent**.¹⁸

4. Pg. 8, L. 10. In light of the above, it would be useful to see a direct comparison of DSBs by PFGE upon infection by Hp *cag*-PAI -negative and -positive strains in Fig. 3 or supp Fig 3, rather than simply referencing the finding (#18).

Response: Yes. We know this is a lame excuse, but our PFGE apparatus broke down during the revision and we simply didn't have the opportunity to run gels with Cag-PAI and RfaE mutant-infected samples side by side. We therefore have to point to the literature for this statement (which is well backed up by published data).

5. Fig. 3. Perhaps I missed this, but DSBs are not shown for CTRL cells supplemented with α ADP-heptose and β ADP-heptose, are they? If not, this should be shown.

Response: This is now shown in a pulsed-field gel and its quantification in a new panel of suppl. Figure 3, and described in the text as follows. We indeed observed fragmented DNA upon addition of the β -ADP-heptose, to an extent that was quite comparable to the live wild type infection.

Results, p.8: The β -anomer, but not the α -anomer, induced DNA damage -as judged by 53BP1 formation- as well as IL-8 secretion in a TIFA/ALPK1-dependent manner (Fig. 3f,g, suppl. Fig. 3b); **which was also confirmed by PFGE of damaged DNA (suppl. Fig. 3c,d).**

6. In lines 5 and 6 of page 8, replace supp. Fig. 3a with 3ab.
In line 8 of page 8, replace supp. Fig 3b with 3c.
In line 15 of page 8, replace supp. Fig 3c with 3b.

Response: These call outs are now all outdated as this supplemental figure has changed a lot; all call outs should now be accurate.

7. Fig. 4C. It appears that there is a further decreased replication stress for Hp WT infection of the TIFA mutant compared with the ALPK1 mutant. Also, in the case of β ADP-heptose addition for these two mutants. Are these statistically significant differences, if so, what are the authors' thoughts? Could there be an additional ALPK1-independent but TIFA-dependent mechanism for promoting DNA replication stress upon Hp infection?

Response: In general, TIFA and ALPK1 deficiency had extremely robust and reproducible effects on all readouts examined here. We followed the suggestion of reviewer 3 and confirmed several results with an independently generated ALPK1 ko cell line in the AGS background, and also were able to complement the TIFA phenotypes with wild type TIFA expressed under the viral MND promoter. In some readouts, the ALPK1 ko had stronger effects and in others the TIFA ko. We don't believe there to be consistently stronger effects of TIFA deficiency over ALPK1 deficiency if all readouts are examined together. Albeit statistically significant, we believe we shouldn't overinterpret the difference in replication stress observed in Figure 4c, as correctly pointed out by this reviewer.

Results, p.5:

The genetic ablation of *ALPK1* or *TIFA* in AGS cells strongly reduced DNA DSBs as judged by quantification of 53BP1 foci (Fig. 1a,b). DNA damage was limited to cells in S-phase, which were identified by PCNA staining (Fig. 1b, suppl. Fig. 1a) and was also observed with a second strain of *H. pylori* (suppl. Fig. 1b). Plotting the signal intensities of PCNA over DAPI, which readily separates cells in G1, S and G2/M phases of the cell cycle, confirmed that cells with five and more 53BP1 foci are typically in S-phase (Fig. 1c). To rule out that off-target effects of genome-editing by CRISPR led to the reduction in 53BP1 foci, we took advantage of a second, independently generated ALPK1-deficient AGS line published previously²² and of *TIFA*-deficient AGS cells that had been complemented by transduction with a lentivirus containing the complete *TIFA* coding sequence.²¹ In these complemented cells, *TIFA* expression is driven by the lentiviral MND promoter and completely restores IL-8 production upon co-culture with *H. pylori*.²¹ When subjected to *H. pylori* exposure followed by 53BP1 staining and the quantification of 53BP1 foci, the second *ALPK1*-deficient cell line phenocopied the effects of the first one (suppl. Fig. 1c); *TIFA*-complemented cells exhibited an almost complete restoration of 53BP1 foci formation, whereas the baseline levels (uninfected condition) were unchanged (suppl. Fig. 1c). These results indicate that the resistance of *ALPK1*- or *TIFA*-deficient AGS cells to *H. pylori*-induced DNA damage is indeed due to *TIFA* ablation and not off-target effects of CRISPR.

and

Results, p.9:

AGS cells lacking *ALPK1* or *TIFA* expression were resistant to replication fork slowing in this assay, both in the setting of live infection and of exposure to β -ADP-heptose (Fig. 4b,c, suppl. Fig. 4a). Complementation of *TIFA*-deficient AGS cells by *TIFA* overexpression from the lentiviral MND promoter rescued the effects of live *H. pylori* infection on fiber shortening (suppl. Fig. 4c).

8. Figs. 4E and F. Addition of TNF α suppressed replication stress significantly, but this is not described in the text. Describe and interpret.

Response: The comparison of untreated to TNF- α treatment in Figure 4f is not statistically significant. TNF- α never really did much in any of the readouts, neither in AGS nor in U2OS cells, and neither in the immunofluorescence readouts of R-loops or 53BP1 foci, nor in the fiber assay. This is now discussed in a bit more detail in the discussion as mentioned above (point 1).

9. Pg. 11, L. 1-3. The “data not shown” should be shown in a supplemental Fig.

Response: This data is now included in suppl. Fig. 6a. There is only background signal in the absence of DOX, and the small nuclear foci indicative of RNase H1 are missing.

10. P. 13, L. 11-12. It is mentioned that the authors identified 17 genes with functions of R-loop prevention and processing through a literature search. These references should be cited.

Response: The appropriate references for all R-loop metabolism genes investigated have now been added to the appropriate section:

Results, p.14: We also looked specifically for mutations and copy number variations (CNVs) in genes involved in R-loop prevention and processing, with the idea that mutations in such genes might predispose to gastric carcinogenesis. We plotted the mutations and CNVs in 18 genes with functions in R-loop prevention or processing relative to seven known gastric cancer driver genes, i.e. *CDH1*, *APC*, *TP53*, *ARID1A*, *PIK3CA*, *KRAS* and *ERBB2*, and relative to the mutational burden per megabase. The 18 genes with R-loop metabolism-related functions were selected based on literature searches: we focused on the RNA/DNA helicases *AQR*, *SETX* and *DHX9* involved in R-loop unwinding,^{19, 43, 44} the DNA helicase *PIF1* known to prevent the accumulation of R-loops at tRNA genes,⁴⁵ the DNA topoisomerase *TOP3B*, which also prevents accumulation of R-loops,⁴⁶ the endonuclease *RNASEH2* which, like RNase H1 cleaves the RNA strand in RNA/DNA hybrids,⁴⁷ the splicing factor *SRSF1*, known to prevent R-loop formation,⁴⁸ *THOC1* and *THOC2*, components of the THO/TREX complex involved in mRNA export and R-loop prevention,⁴⁹ the THO/TREX complex associated histone deacetylase *SIN3A*,⁵⁰ and the six components of the TREX-2 complex that, with *BRCA2*, is also associated with mRNA export and R-loop prevention.⁵¹ Several of these genes were recurrently (in up to 36% of gastric cancer cases) affected by CNVs and/or missense or frameshift mutations, with the most recurrently affected genes belonging to the TREX-2 complex (*ENY2*, *MCM3AP*, *SEM1*, *CETN3*) or having RNA/DNA helicase or endonuclease activity (*SETX*, *AQR*, *DHX9*, *RNASEH2*) (suppl. Fig. 7c). Mutations in R-loop metabolism genes were largely predicted to be damaging mutations, and -not unexpectedly- were mostly detected in the microsatellite-unstable subtype of gastric cancer that exhibited the highest overall mutational burden; CNVs affecting R-loop genes were mostly detected in the chromosomally unstable subtype (suppl. Fig. 7c). In conclusion, the mutational landscape of gastric cancer provides circumstantial evidence for the hypothesis that *H. pylori* infection favors chromosomal instability, and that the inactivation of genes encoding helicases, RNases and other factors involved in R-loop prevention or processing may be an early event in gastric carcinogenesis, predisposing *H. pylori*-infected cells to aberrant R-loop accumulation and its potentially deleterious consequences.

11. P. 16, L. 17-21. Hp infection is linked to genomic instability via reactive oxygen and nitrogen species reviewed by Kidane D (International Journal of Molecular Sciences 2018, 19:2891). Although the authors mention ways by which R-loops might be formed upon Hp infection, they do not experimentally address or entertain the possibility that damaged nucleotides by reactive oxygen and nitrogen species contribute to R-loops. It is conceivable that replication fork blockage at lesion sites in the sense strand inflicted by reactive oxygen/nitrogen species can cause the accumulation of transcripts from the antisense strand and lead to formation of R-loops.

Response: This is a very good point raised by our reviewer. We have now addressed the role of ROS in R-loop-induced DNA damage and fork slowing in detail. We can confirm what was reported in Kidane et al (a paper that is now also cited), namely that ROS form in *H. pylori*-infected cells. Treatment of infected cells with the antioxidant N-acetyl-cysteine blocked ROS formation but not DNA damage or fork slowing as assessed in multiple replicate experiments. This data is now shown in suppl. Figures 2 and 4, and explained in the following amendments to the text:

Results, p.6:

All results combined indicate that activation of the ALPK1/TIFA/NF- κ B signaling axis by *H. pylori* triggers DNA damage in target cells that appears to be specific to this pathway of NF- κ B activation, and that requires active transcription and occurs predominantly in actively replicating (S-phase) cells. As NF- κ B activation and the resulting production of reactive oxygen species (ROS), through mechanisms involving inducible nitric oxide synthase (iNOS) and other inflammatory enzymes, have been implicated in *H. pylori*-induced DNA damage,²⁹ we addressed this possibility experimentally using the antioxidant N-acetyl-cysteine. We found that *H. pylori* induces ROS, as judged by their flow cytometric detection and quantification (suppl. Figure 2b). However, co-culturing AGS cells with *H. pylori* in the presence of N-acetyl-cysteine -at concentrations that completely abrogated ROS production- failed to reduce the DNA damage as judged by 53BP1 foci formation (suppl. Figure 2b-d). This result argues against a major contribution of ROS to *H. pylori*-induced DNA damage in this setting.

Results, p.10:

The slowing of replication forks upon *H. pylori* infection was dependent on RfaE and the Cag-PAI, as both null mutants failed to cause CldU tract shortening (Fig. 4e,f). TNF- α exposure did not cause CldU tract shortening (Fig. 4e,f), and also failed to rescue the phenotype of the RfaE mutant (suppl. Fig. 4d). Exposure of infected cells to the NF- κ B inhibitor BAY 11-7082 or to the transcription inhibitor triptolide rescued *H. pylori*-induced fork slowing (Fig. 4e,f). In contrast, treatment with the antioxidant N-acetyl-cysteine did not prevent fork slowing (suppl. Fig. 4d). The combined results suggest that active transcription driven by NF- κ B as a consequence of β -ADP-heptose delivery and ALPK1/TIFA signaling is a prerequisite of replication fork slowing in *H. pylori*-infected cells; in contrast, ROS produced by infected cells do not contribute to DNA damage or replication stress.

Reviewer #3 (Expertise: Gastric cancer, H.Pylori, Remarks to the Author):

Bauer et al. show that *H. pylori* generates DSBs (53BP1 foci) in S-phase cells by activating the NF- κ B transcription factor in a manner that is dependent on the T4SS/ALPK/TIFA/NF- κ B signaling axis, which was recently identified by several groups including the authors' group (Zimmermann et al. Cell Reports 20, 2384-2395, 2017; Gall et al. mBio 8, e011687-17, 2017; Stein et al. PLoS Pathog 13, e1006514, 2017). The authors also show that *H. pylori* infection-mediated DSBs are due to accelerated R-loop formation in NF- κ B-regulated genes, which induces replication fork stress or by cutting the R-loop with the nucleotide excision repair enzymes such as XPG or XPF as reported previously by the authors' group (Hartung et al. Cell Reports 13, 70-79, 2015).

Although the experiments were well-designed and the results were nicely demonstrate, my overall impression after reading the manuscript is that it is made by putting together two works previously published by the authors' group [(Hartung et al. Cell Reports 13, 70-79, 2015) and (Zimmermann et al. Cell Reports 20, 2384-2395, 2017)] by arguing that increase in transcriptional activity promotes R-

loop formation and subsequent replication stress. Indeed, it has already been shown that the nucleotide excision repair endonucleases XPG and XPF promote R-loop formation and thereby induce DSBs. Since most of the molecular mechanisms involved in DSB induction following *H. pylori* infection presented in this work were reported or discussed in their previous works (Hartung et al. 2015 and by Zimmermann et al. 2017), I cannot find any novel findings/discoveries or new conceptual breakthroughs in the present study.

Response: We are of the strong opinion that one cannot possibly conclude from previously published data that *H. pylori* infection induces R-loop-mediated replication stress (the key finding of this work).

The title of the paper “The ALPK1/TIFA/NF- κ B axis links a bacterial carcinogen to R-loop-induced replication stress” argues that their finding is associated with the cancer development. Although genomic instability has been shown to play an important role in carcinogenesis, the present study does not describe how it could promote gastric carcinogenesis by *H. pylori* (including the merit of cancer development for *H. pylori*)? Especially, host gastric epithelial cells receiving massive DNA damages (DSBs) should undergo apoptosis and thus excluded from the body unless repaired by an extremely efficient way. If the *H. pylori*-damaged cells could survive despite severe DSBs, why and how? I think that this question needs to be clarified in order to argue the relationship between *H. pylori*-induced DNA damage and carcinogenesis.

Response: This topic now receives the attention it deserves in the revised version. It is absolutely true that cells undergoing such massive DNA damage should launch a DDR and ultimately die. However, we have noticed again and again that infected cells survive DESPITE the DNA damage, and now quantify and show this in Figure 7. Also, there are no signs of a DDR in infected cells, which clearly fail to activate ATM. Furthermore, we now show that *H. pylori* infection leads to micronuclei formation (Fig. 7a-d). These small, extranuclear chromatin bodies can be reincorporated into the daughter cell nucleus and contribute to genomic instability, a hallmark of cancer. And finally, we show (see also point 3 below) that the presence of *H. pylori* is associated with a higher mutational burden in gastric cancer, which further provides (circumstantial) evidence for the pathogen inflicting DNA damage and genomic instability in the cancer that it causes.

It is also weird why activation of NF- κ B other than the non-ALPK1/TIFA axis is incapable of inducing DSBs in the host cells. The finding suggests that activation of NF- κ B is required but not sufficient for induction of DSBs and suggests the involvement of additional *H. pylori*-host cell interaction. This possibility could be tested by treatment of AGS cells that had been infected with *H. pylori* strains lacking RfaE or T4SS with TNF α .

Response: This is a great suggestion and has now been done, with the assumption that TNF- α would provide the NF- κ B stimulus that is lacking upon exposure to the RfaE mutant. However, in multiple experiments, we could not find evidence of a rescue of the RfaE mutant defect with TNF- α , suggesting that the specific activation of NF- κ B via the ALPK1/TIFA pathway is required for DNA damage, and further indicating that not simply ANY kind of NF- κ B activation is sufficient to generate DNA damage.

This is now shown in suppl. Figure 3f, along with the following text:

Results, p.8: As in the context of live *H. pylori* infection, DNA damage upon β -ADP-heptose exposure was limited to PCNA⁺ cells (Fig. 3f, suppl. Fig. 3b,e). **The defect of the *rfaE* mutant could not be rescued by addition of TNF- α (suppl. Fig. 3f).** The combined results indicate that *rfaE* activity is required, and its product β -ADP-heptose is sufficient, to induce the ALPK1/TIFA-dependent DNA damage observed upon live *H. pylori* infection of cultured gastric epithelial cells.

The followings are other specific comments:

1. In Figs. 1, 3, and 4, the authors used ALPK1- or TIFA-knockout AGS cells established by a CRISPR/Cas9 technology. The each knockout cell line was generated by using a single guide RNA sequence. To rule out off-target effects, multiple targeting sequences are required for the same target. The authors should reproduce the data using multiple knockout cell lines.

Response: This is a great suggestion! We have now used two approaches to deal with this request. On the one hand, we used a second, independently generated ALPK1 ko cell line (generated by the Meyer lab), which phenocopied the effects of ALPK1 ko as seen in the first such cell line used. On the other hand, we used a complementation strategy to rule out off-target effects of TIFA genome editing. The TIFA-deficient AGS cells were complemented by transduction with a lentivirus containing the complete TIFA coding sequence. This complemented cell line had been shown in the original publication by our co-authors Tina Gall and Nina Salama to exhibit complete restoration of IL-8 production in response to *H. pylori* infection (Figure 2B, Gall et al MBio 2017). TIFA expression in the complemented cells is driven by the lentiviral MND promoter.

The complemented cell line has now been tested in three independent rounds of experiments each in the DNA damage assay, i.e. immunofluorescent staining for 53BP1 and gH2AX, and the DNA fiber assay. In both settings, TIFA complementation restored the DNA damage and replication fork slowing to wild type AGS levels.

The second ALPK1 ko cell line has been used for immunofluorescent staining for 53BP1 and gH2AX; these results of the ALPK1 ko and complemented TIFA cell lines are shown in suppl. Figures 1 and 4 along with the following modified text.

Results, p.5:

The genetic ablation of *ALPK1* or *TIFA* in AGS cells strongly reduced DNA DSBs as judged by quantification of 53BP1 foci (Fig. 1a,b). DNA damage was limited to cells in S-phase, which were identified by PCNA staining (Fig. 1b, suppl. Fig. 1a). Plotting the signal intensities of PCNA over DAPI, which readily separates cells in G1, S and G2/M phases of the cell cycle, confirmed that cells with five and more 53BP1 foci are typically in S-phase (Fig. 1c). **To rule out that off-target effects of genome-editing by CRISPR led to the reduction in 53BP1 foci, we took advantage of a second, independently generated ALPK1-deficient AGS line published previously²² and of TIFA-deficient AGS cells that had been complemented by transduction with a lentivirus containing the complete TIFA coding sequence.²¹ In these complemented cells, TIFA expression is driven by the lentiviral MND promoter and completely restores IL-8 production upon co-culture with *H. pylori*.²¹ When subjected to *H. pylori* exposure followed by 53BP1 staining and the quantification of 53BP1 foci, the second *ALPK1*-deficient cell line phenocopied the effects of the first one (suppl. Fig. 1c); TIFA-complemented cells exhibited an almost complete restoration of 53BP1 foci formation, whereas the baseline levels (uninfected**

condition) were unchanged (suppl. Fig. 1c). These results indicate that the resistance of *ALPK1*- or *TIFA*-deficient AGS cells to *H. pylori*-induced DNA damage is indeed due to *TIFA* ablation and not off-target effects of CRISPR.

and

Results, p.9:

AGS cells lacking *ALPK1* or *TIFA* expression were resistant to replication fork slowing in this assay, both in the setting of live infection and of exposure to β -ADP-heptose (Fig. 4b,c, suppl. Fig. 4a). Complementation of *TIFA*-deficient AGS cells by *TIFA* overexpression from the lentiviral MND promoter rescued the effects of live *H. pylori* infection on fiber shortening (suppl. Fig. 4c).

2. In Fig. 2d, they should show that γ H2AX and 53BP1 staining relative to *H. pylori* staining in serial sections.

Response: We spent several weeks trying to get the γ H2AX staining going for gastric specimens, but despite trying out multiple antibody concentrations and playing with the incubation time, we were never able to obtain a signal in the gastritis that was truly specific (and stronger than in the control mucosa). We (i.e our co-author Prof. Achim Weber, the pathologist working with us on this problem) have terrific experience with the same γ H2AX antibody on liver tissue, but the background problem in the stomach sadly precluded showing any of this data. We felt that the background observed in normal gastric mucosa precluded drawing any strong conclusions with respect to DNA damage in vivo, and felt we had to limit ourselves to the statement that *H. pylori* resides in the vicinity of replicating cells. This observation is clearly backed by the data.

3. In Fig. 7c, they analyzed the mutations of the R-loop genes in the gastric cancer genomic data reported by the Cancer Genome Atlas (TCGA). However, they did not show that correlation between the mutations and *H. pylori* infection.

Response: This whole section has now been revised thoroughly. We have moved the analysis of R-loop gene mutations to the supplement, and have instead, in main Figure 7, included a heatmap of genes that are differentially mutated as a function of *H. pylori* status. To this end, we annotated the TCGA data with their *H. pylori* status based on the presence of *H. pylori* transcripts and genomic sequences among non-human sequences detected in the TCGA omics data. This analysis was informative indeed, and has been included in the text as follows:

Results, p.13:

We next asked whether gastric cancer genomic data would possibly provide circumstantial evidence for the presence of *H. pylori* as a driving force of malignant transformation. To this end, we used the non-human transcript and genomic information that is present in publicly accessible multi-omics datasets of gastric cancer, which had been generated as part of the Cancer Genome Atlas (TCGA) project, to annotate gastric cancer samples with their *H. pylori* status. We had access to multi-omics datasets for 291 tumors from treatment-naïve gastric cancer patients that had previously been subjected to array-based somatic copy number analysis, whole-exome sequencing and array-based DNA methylation profiling, and that had resulted in the description of four major genetically defined subtypes of gastric cancer.⁴¹ These subtypes were EBV-positive gastric cancer (characterized by presence of EBV genes and transcripts, a very low mutational burden, *PIK3CA* mutations and *CDKN2A* hypermethylation), microsatellite-unstable gastric cancer (high mutational burden, MSI high, MLH1

hypermethylation), chromosomally unstable gastric cancer (with large numbers of copy number variations and a high mutational burden, and ubiquitous TP53 mutations) and genomically stable gastric cancer (low mutational burden, diffuse type by histology, early onset, *RHOA* and *ARHGAP6/26* somatic genomic alterations).⁴¹ We were able to recapitulate the stratification of the 291 patients into these subtypes (Fig. 7e). We found evidence of *H. pylori* presence in all four subtypes, with 1/3 to 1/2 of patients of each subtype exhibiting evidence of *H. pylori* transcripts or genomic DNA in either their tumor and/or adjacent tissue (Fig. 7e). When comparing the tumor mutational burden of the four subtypes, we found that tumors from patients with evidence of *H. pylori* infection (in either tumor and/or adjacent tissue) of the most common MSI and CIN subtypes had a higher mutational burden than those without *H. pylori* infection: within the CIN subtype, which with 136 samples was the most common of the four gastric cancer subtypes in the TCGA dataset, 57 *H. pylori*-infected patients had on average 2 +/- 1.5 mutations per megabase, whereas this number was 1.54 +/- 0.98 in the 79 *H. pylori*-negative patients ($p=0.052$ as determined by student's T-test). Within the MSI subtype the figures were 20 +/- 16 for 23 *H. pylori*-positive vs. 16 +/- 7 for 33 *H. pylori*-negative patients, but the difference was not statistically significant. ($p=0.2$). A total of 232 genes were differentially affected by mutations as a function of *H. pylori* infection across all four subtypes (suppl. table 1, Fig. 7e).

REVIEWERS' COMMENTS:

Reviewer#1: (Remarks to the Author)

The authors have made a great job revising the manuscript. It is now much stronger and conclusions are better supported and convincing. I think it is appropriate for Nat Comm in its present form.

Reviewer#3: (Remarks to the Author)

As stated previously, the authors' group showed that infection of gastric epithelial cells with *H. pylori* induces DSBs by activating the NF- κ B transcription factor in a manner that is dependent on the T4SS/ALPK/TIFA/NF- κ B signaling axis (Zimmermann et al. Cell Reports, 2017). They also showed that *H. pylori* infection-mediated DSBs are due to accelerated R-loop formation in NF- κ B-regulated genes, which induces replication fork stress or by cutting the R-loop with the nucleotide excision repair enzymes such as XPG or XPF (Hartung et al. Cell Reports, 2015). Taken these together, it is quite reasonable to speculate that *H. pylori* infection induces R-loop-mediated replication stress through the T4SS/ALPK/TIFA/NF- κ B signaling axis. Of course, I do not deny the authors' argument that "We are of the strong opinion that one cannot possibly conclude from previously published data that *H. pylori* infection induces R-loop-mediated replication stress." In this regard, the authors conducted new experiments in response to my comment and found that the specific activation of NF- κ B via the ALPK1/TIFA pathway is required for DNA damage. Does this mean that R-loop formation, the key finding of this paper according to the authors, is not a pivotal event for the *H. pylori*-mediated DSB induction as it should also be induced by TNF α -mediated NF- κ B activation?

In response to my specific comment 3, the authors provided new data using the non-human transcript and genomic information from the TCGF database to annotate gastric cancer samples with their *H. pylori* infection status. They found that 1/3 to 1/2 (33%~50%) of gastric cancer patients exhibiting evidence of *H. pylori* transcripts or genomic DNA in either their tumor and/or adjacent tissue. They then found that the MSI and CIN subtypes of gastric cancers from patients with evidence of *H. pylori* infection had a higher mutational burden than those without *H. pylori* infection. With regard to this, it has already been shown that more than 80% of gastric cancers are associated with *H. pylori* infection. Also notably, *H. pylori* disappear from the stomach with severe intestinal metaplasia (the precancerous gastric lesion) caused by long-term infection with *H. pylori*. A big question with their TCGA study is therefore whether the presence or absence of the *H. pylori* genomes (or transcripts) in the database faithfully reflects the status of *H. pylori* infection for each gastric cancer patients (i.e., a large number of false-negative cases, which substantially influence the results of their statistical analysis,

may exist). This point needs to be clarified to justify their conclusions. Be that as it may, the results obtained by the analysis of the TCGA data do not provide direct pathophysiological link between their work and the development of gastric cancer in terms of genomic instability.

Finally, the authors did not satisfactorily responded to my question as to why *H. pylori*-infected cells survive despite suffering from severe DNA damages. Obviously, presence of extracellular chromatin bodies does not give any clue to the question.

Reviewer#4: (Remarks to the Author, replacement for Reviewer#2)

Long-term infections with the bacterial type-I carcinogen *Helicobacter pylori* have been associated with a broad range of gastric disorders, including gastritis, ulceration, gastric cancer or MALT lymphoma. Infection of gastric epithelial cells with the *H. pylori* causes DNA double strand breaks in the host cell chromosome. Here, the authors showed that *H. pylori*-triggered DNA damage occurs co-transcriptionally in S-phase cells that activate NF- κ B signaling upon innate immune recognition of the LPS metabolite ADP-heptose through the ALPK1/TIFA signaling cascade published recently. It appears that DNA damage depends on the *rfaE* and *cag* pathogenicity island genes of *H. pylori*, which is accompanied by replication fork stalling. This has also been convincingly demonstrated in cells derived from gastric organoids. Remarkably, *H. pylori*-induced replication stress and DNA damage depend on the presence of co-transcriptional RNA/DNA-hybrids (R-loops) that arise during S-phase as a consequence of stimulated ADP-heptose/ALPK1/TIFA pathway. Also, *H. pylori* appears to reside in close proximity to S-phase cells in the gastric mucosa of gastritis patients. Taken together, the presented data link nicely bacterial infection and NF- κ B-driven innate immune responses to R-loop-dependent replication stress and DNA damage caused by *H. pylori*. This is highly interesting and well-controlled paper, which will raise considerable attention in the community.

I have been asked very recently to serve as adhoc reviewer after the revision and to replace the second reviewer, who was not available anymore. Therefore, I have read the comments and expectations by all previous reviewers and the corresponding rebuttal letter by the authors. Overall, I think that the authors made a very nice job and answered comprehensively in the text and with additional experiments, which appear more than sufficient to me. Thus, I believe that the revised paper is in its final stage. I have no further experimental requests for improvement.

REVIEWERS' COMMENTS: (answers in red)

Reviewer#1: (Remarks to the Author)

The authors have made a great job revising the manuscript. It is now much stronger and conclusions are better supported and convincing. I think it is appropriate for Nat Comm in its present form.

Thank you for this assessment!

Reviewer#3: (Remarks to the Author)

As stated previously, the authors' group showed that infection of gastric epithelial cells with *H. pylori* induces DSBs by activating the NF- κ B transcription factor in a manner that is dependent on the T4SS/ALPK/TIFA/NF- κ B signaling axis (Zimmermann et al. Cell Reports, 2017). They also showed that *H. pylori* infection-mediated DSBs are due to accelerated R-loop formation in NF- κ B-regulated genes, which induces replication fork stress or by cutting the R-loop with the nucleotide excision repair enzymes such as XPG or XPF (Hartung et al. Cell Reports, 2015). Taken these together, it is quite reasonable to speculate that *H. pylori* infection induces R-loop-mediated replication stress through the T4SS/ALPK/TIFA/NF- κ B signaling axis. Of course, I do not deny the authors' argument that "We are of the strong opinion that one cannot possibly conclude from previously published data that *H. pylori* infection induces R-loop-mediated replication stress.". In this regard, the authors conducted new experiments in response to my comment and found that the specific activation of NF- κ B via the ALPK1/TIFA pathway is required for DNA damage. Does this mean that R-loop formation, the key finding of this paper according to the authors, is not a pivotal event for the *H. pylori*-mediated DSB induction as it should also be induced by TNF α -mediated NF- κ B activation?

R-loops are NOT induced by TNF- α , but seem to indeed be a rather pivotal event in NF- κ B activation by the ALPK1/TIFA signaling axis. Also, combining TNF- α with infection using an RfaE mutant did not lead to R-loops (all shown in supplemental figures), indicating that the ADP-heptose, and not other factors of *H. pylori* or NF- κ B activation through the canonical pathway are alone or in combination sufficient to induce R-loops and DNA damage.

In response to my specific comment 3, the authors provided new data using the non-human transcript and genomic information from the TCGF database to annotate gastric cancer samples with their *H. pylori* infection status. They found that 1/3 to 1/2 (33%~50%) of gastric cancer patients exhibiting evidence of *H. pylori* transcripts or genomic DNA in either their tumor and/or adjacent tissue. They then found that the MSI and CIN subtypes of gastric cancers from patients with evidence of *H. pylori* infection had a higher mutational burden than those without *H. pylori* infection. With regard to this, it has already been shown that more than 80% of gastric cancers are associated with *H. pylori* infection. Also notably, *H. pylori* disappear from the stomach with severe intestinal metaplasia (the precancerous gastric lesion) caused by long-term infection with *H. pylori*. A big question with their TCGA study is therefore whether the presence or absence of the *H. pylori* genomes (or transcripts) in the database faithfully reflects the

status of *H. pylori* infection for each gastric cancer patients (i.e., a large number of false-negative cases, which substantially influence the results of their statistical analysis, may exist). This point needs to be clarified to justify their conclusions. Be that as it may, the results obtained by the analysis of the TCGA data do not provide direct pathophysiological link between their work and the development of gastric cancer in terms of genomic instability.

We certainly don't pretend that the TCGA data provide a direct pathophysiological link between R-loop induced DNA damage and gastric carcinogenesis. That would indeed be preposterous. Rather, we state that the TCGA data provides circumstantial evidence that *H. pylori* is associated with MORE mutations and a different mutation pattern in the MSI and CIN subtypes of gastric cancer. Not more and not less. We are now, following our editors suggestion, further toning down our already quite weak statements on the TCGA data, by modifying the following paragraph in the discussion section (new text in red):

Discussion, p.19: In agreement with the hypothesis that *H. pylori*-induced DNA damage causes genotoxicity and predisposes to mutagenesis, we find more mutations, and different mutation patterns, in gastric cancer samples of patients with evidence of *H. pylori* presence at the time of diagnosis. As the diagnosis of *H. pylori* presence in non-human sequences of gastric cancer and adjacent tissue is prone to false-negative (not so much false-positive) results, the differences observed in the mutational burden need to be interpreted with caution; this is especially true because *H. pylori* disappears from its gastric niche as intestinal metaplasia and other gastric cancer precursor lesions form. We propose that inactivating mutations or copy number losses in genes involved in the prevention or elimination of R-loops are early events in gastric cancer that predispose hyper-proliferating cells to *H. pylori*-induced R-loop accumulation and DNA damage.

Finally, the authors did not satisfactorily responded to my question as to why *H. pylori*-infected cells survive despite suffering from severe DNA damages. Obviously, presence of extracellular chromatin bodies does not give any clue to the question.

They survive because they avoid triggering a DNA damage response (now shown by Western blotting for ATM activation, and phosphorylation of its downstream target KAP1).

Reviewer#4: (Remarks to the Author, replacement for Reviewer#2)

Long-term infections with the bacterial type-I carcinogen *Helicobacter pylori* have been associated with a broad range of gastric disorders, including gastritis, ulceration, gastric cancer or MALT lymphoma. Infection of gastric epithelial cells with the *H. pylori* causes DNA double strand breaks in the host cell chromosome. Here, the authors showed that *H. pylori*-triggered DNA damage occurs co-transcriptionally in S-phase cells that activate NF- κ B signaling upon innate immune recognition of the LPS metabolite ADP-heptose through the ALPK1/TIFA signaling cascade published recently. It appears that DNA damage depends on the *rfaE* and *cag* pathogenicity island genes of *H. pylori*, which is accompanied by replication fork stalling. This has also been convincingly demonstrated in cells derived from gastric organoids. Remarkably, *H. pylori*-induced replication stress and DNA damage depend on the presence of co-transcriptional RNA/DNA-hybrids (R-loops) that arise during S-phase as a consequence of stimulated ADP-heptose/ALPK1/TIFA pathway. Also, *H. pylori* appears to

reside in close proximity to S-phase cells in the gastric mucosa of gastritis patients. Taken together, the presented data link nicely bacterial infection and NF- κ B-driven innate immune responses to R-loop-dependent replication stress and DNA damage caused by *H. pylori*. This is highly interesting and well-controlled paper, which will raise considerable attention in the community.

I have been asked very recently to serve as adhoc reviewer after the revision and to replace the second reviewer, who was not available anymore. Therefore, I have read the comments and expectations by all previous reviewers and the corresponding rebuttal letter by the authors. Overall, I think that the authors made a very nice job and answered comprehensively in the text and with additional experiments, which appear more than sufficient to me. Thus, I believe that the revised paper is in its final stage. I have no further experimental requests for improvement.

Thanks also to this reviewer for replacing reviewer 2 and for being positive and enthusiastic about our data.